**Article**  https://doi.org/10.1038/s41467-023-37937-4

# Patterns in soil microbial diversity across Europe

Maëva Labouyrie [1,2,3], Cristiano Ballabio[2], Ferran Romero [3], Panos Panagos [2], Arwyn Jones[2], Marc W. Schmid [4], Vladimir Mikryukov[5,6], Olesya Dulya[5,6], Leho Tedersoo [5], Mohammad Bahram [6,7], Emanuele Lugato [2], Marcel G. A. van der Heijden [1,3] ✉ & Alberto Orgiazzi [2] ✉

Factors driving microbial community composition and diversity are well established but the relationship with microbial functioning is poorly understood, especially at large scales. We analysed microbial biodiversity metrics and distribution of potential functional groups along a gradient of increasing land-use perturbation, detecting over 79,000 bacterial and 25,000 fungal OTUs in 715 sites across 24 European countries. We found the lowest bacterial and fungal diversity in less-disturbed environments (woodlands) compared to grasslands and highly-disturbed environments (croplands). Highly-disturbed environments contain significantly more bacterial chemoheterotrophs, harbour a higher proportion of fungal plant pathogens and saprotrophs, and have less beneficial fungal plant symbionts compared to woodlands and extensively-managed grasslands. Spatial patterns of microbial communities and predicted functions are best explained when interactions among the major determinants (vegetation cover, climate, soil properties) are considered. We propose guidelines for environmental policy actions and argue that taxonomical and functional diversity should be considered simultaneously for monitoring purposes.

Soil biota plays an important role in contributing to provide ecosystem services such as food production, climate regulation and pest control[1,2]. Soil microbes are involved in the decomposition of soil organic matter, regulate carbon stocks and nutrient cycling, and facilitate plant nutrient uptake[3,4]. Changes in soil microbial community composition and related functions can alter the associated services[5]. While previous studies have shown that land-use perturbation can significantly reduce above-ground biodiversity[6], and that vegetation cover, climate and soil properties can strongly affect above-ground communities[7], far less is known about the impacts of anthropogenic and environmental factors on below-ground diversity and functions, especially at large spatial (e.g. continental) scales[8].

At the community-level, land-use perturbation has been identified as one of the main anthropic pressures affecting soil microbial diversity, resulting in community composition shifts[8,9]. Other reported factors influencing soil microbial communities are climate, soil properties and vegetation. Changes in climate and in specific soil properties, such as pH[10,11], texture[12] and nitrogen availability[13] also lead to changes in microorganism assemblages[14]. In addition, plant community attributes and functional traits predict a unique portion of the variation in soil microbial diversity and community composition,

[1]Department of Plant and Microbial Biology, University of Zurich, Zürich, Switzerland. [2]European Commission, Joint Research Centre (JRC), Ispra, VA, Italy. [3]Plant-Soil-Interactions, Research Division Agroecology and Environment, Agroscope, Zürich, Switzerland. [4]MWSchmid GmbH, Glarus, Switzerland. [5]Mycology and Microbiology Center, University of Tartu, Tartu, Estonia. [6]Department of Botany, Institute of Ecology and Earth Sciences, University of Tartu, Tartu, Estonia. [7]Department of Ecology, Swedish University of Agricultural Sciences, Uppsala, Sweden. ✉e-mail: marcel.vanderheijden@agroscope.admin.ch; alberto.orgiazzi@gmail.com

which could not be explained by soil physico-chemical properties and climate[15].

At the functional level, land-use perturbation may alter the composition of soil functional groups (e.g. mycorrhizal fungi)[8], by affecting plant diversity and altering carbon and nitrogen retention[16,17]. Moreover, climate, soil properties and vegetation are known to strongly affect the potential functions provided by the microbial communities. Climate warming and altered humidity have been shown to enhance the presence of certain functional groups, such as plant pathogens[18,19], but reduce the abundance of others, such as beneficial arbuscular mycorrhizal fungi[20]. Soil microbial functional groups differ between land-use types (e.g. agricultural and forest soils) and within the same vegetation cover type (e.g. broadleaved and coniferous forests)[21]. At the same time, soil physico-chemical properties (e.g. pH, carbon and nitrogen contents) can significantly shape the distribution of functional groups involved in carbon and nitrogen cycling[22].

Although previous studies have focused on major determinants of soil microbial communities and functional groups, only a few surveys have been conducted at the continental-scale[4,5,9], preventing a systematic assessment of the forces driving changes in taxonomic and functional diversity of bacterial and fungal soil assemblages over large areas. Towards that aim, extensive field studies and standardised datasets accounting for spatial diversity of microbes and driving factors are necessary to better understand the links between microbial assembly distribution and above-mentioned driving factors[23]. Moreover, analyses comparing different microbial domains (e.g. prokaryotes and eukaryotes) are still rarely conducted[24–27] and, generally, do not include both semi-natural (e.g. woodlands and grasslands) and highly-managed (e.g. croplands) environments. In particular, none of the published studies so far has explored and compared bacterial and fungal functional groups among this full range of vegetation cover types and associated land-uses, preventing an assessment of the impacts of an increasing land-use perturbation on targeted functions associated to soil microorganisms. Finally, while many studies have considered driving factors as acting separately, and a few have explored the impacts of their combinations on microbial communities[4,28], the interaction-effect of diverse driver types has, until now, been largely overlooked, especially for microbial functional groups. Our analysis aimed to fill these knowledge gaps.

In this work, we analyse DNA sequences of two groups of microorganisms (bacteria and fungi) from 715 soil samples collected from 23 countries of the European Union and the United Kingdom (EU + UK) (Fig. 1a). We assess continental-scale effects of vegetation cover (and associated land-use), soil properties, climate and their two-way interactions on soil microbial communities and potential functional groups (i.e. inferred based on taxonomy). Soil sampling is performed within the framework of the soil module of the Land Use/Cover Area frame Survey (LUCAS)[29]. Sampling locations cover both semi-natural and highly-managed environments, including six vegetation cover types subjected to an increasing land-use perturbation gradient: from coniferous and broadleaved woodlands, to extensively- and intensively-managed grasslands, and permanent and non-permanent croplands (Fig. 1b). Vegetation cover is combined with a broad set of 9 soil physico-chemical properties and 6 climatic variables. We hypothesise that (i) land-use perturbation affects soil microbial diversity and community structure and potential functional groups, as observed for the above-ground biodiversity[6]; (ii) bacterial and fungal guilds and potential functional groups are shaped by different forces[30,31]; (iii) interactions between vegetation cover, climate and soil properties are more important and informative than single-effects in driving the assembly of bacterial and fungal soil communities and potential functional groups.

Here, we find that microbial diversity and potential functional groups distribution vary along a gradient of increasing land-use perturbation across Europe. Soils that harbour richer and more diverse microbial communities (e.g. croplands and grasslands), also exhibit a higher fraction of potential fungal pathogens. In contrast, woodlands and extensive grasslands harbour more fungal plant symbionts and N-fixing bacteria. Furthermore, vegetation cover, soil properties and climate differently influence bacterial and fungal communities and potential functional groups. In addition, our analysis demonstrates that interactions, more than single environmental factors, drive soil microbial communities and functional groups. Based on these findings, we propose possible environmental policy actions for better preserving soil microbial communities and promoting ecosystem services provided by microbial functional groups.

## Results
### Land-use perturbation effects
The entire dataset included 79,593 bacterial zero-radius operational taxonomic units (zOTUs) and 25,962 fungal OTUs. Bacterial observed zOTU richness and Shannon diversity index were lowest in woodlands and significantly higher in croplands and grasslands. Similarly, fungal richness and diversity were lower in woodlands compared to grasslands and croplands (Fig. 2a–d, Supplementary Table 1). Vegetation cover had a significant impact on bacterial and fungal community structure (β-diversity) (Fig. 2e, f). The highest difference in community structure was found between microbial communities in croplands and woodlands, as supported by pairwise multiple comparison that displayed the highest determination coefficients ($R^2$ = 22.6%; $F$ value = 123.26; $p$ value < 0.001 for bacteria and $R^2$ = 8.3%; $F$ value = 38.05; $p$ value < 0.001 for fungi).

The distribution of inferred bacterial and fungal functional groups also differed among vegetation cover types. Bacterial chemoheterotrophs (12,786 zOTUs, i.e. 16.1% of the total number of bacterial zOTUs) dominated in croplands and intensive grasslands (Fig. 3a). N-fixing bacteria (97 zOTUs, 0.12%) were more abundant in woodlands, as well as in extensive grasslands (Fig. 3b). N-fixing bacteria included both free-living (e.g. *Telmatospirillum siberiense*) and symbiotic organisms (e.g. *Mesorhizobium loti*). Bacterial pathogens (133 zOTUs, 0.17%, among which *Mycobacterium celatum* and *Bacillus anthracis*) dominated in coniferous forests (Fig. 3c). Regarding fungi, ectomycorrhizal symbionts (2361 OTUs, i.e. 9.1% of the total number of fungal OTUs) dominated in woodlands, especially in coniferous forests (Fig. 3d). Arbuscular mycorrhizal fungi (AMF; 1261 OTUs, 4.9%) were more abundant in grasslands, particularly in extensively-managed ones (Fig. 3e). Fungal saprotrophs (9903 OTUs, 35%) and plant pathogens (4355, 16.8%, among which *Fusarum solani*) were dominant in croplands and grasslands (Fig. 3f, g). A total of 65,206 bacterial zOTUs and 7642 fungal OTUs were not assigned to any functional group (even outside previously mentioned groups of interest), representing 81.9% of all bacterial zOTUs and 29.4% of all fungal OTUs.

At the community-level, 23 phyla and 64 classes were identified for bacteria, among which Proteobacteria (30.6%), Actinobacteria (26.1%) and Acidobacteria (22.9%) were the most abundant phyla (Supplementary Table 2). Bacterial classes were dominated by Actinobacteria (25.4%) and Alphaproteobacteria (17.4%) (Supplementary Table 2). For fungi, 20 phyla and 75 classes were identified, among which Ascomycota (49.3%) and Basidiomycota (34.1%) were the most abundant phyla (Supplementary Table 2). Fungal classes were dominated by Agaricomycetes (28.7%), Sordariomycetes (16.7%), Leotiomycetes (10.5%) and Dothideomycetes (10.2%). A small fraction (0.48%) of the fungal OTUs could not be assigned to specific fungal phyla.

Mean relative abundances of Actinobacteria (phylum and class), Ascomycota, Sordariomycetes and Dothideomycetes increased from woodlands to grasslands and croplands (Supplementary Figs. 1, 2a, d, 3a, d, f). On the opposite, Acidobacteria, Alphaproteobacteria, Basidiomycota, Agaricomycetes and Leotiomycetes decreased along those vegetation cover types (Supplementary Figs. 1, 2a, e, 3b, c, e). Proteobacteria dominated in woodlands, particularly in coniferous forests

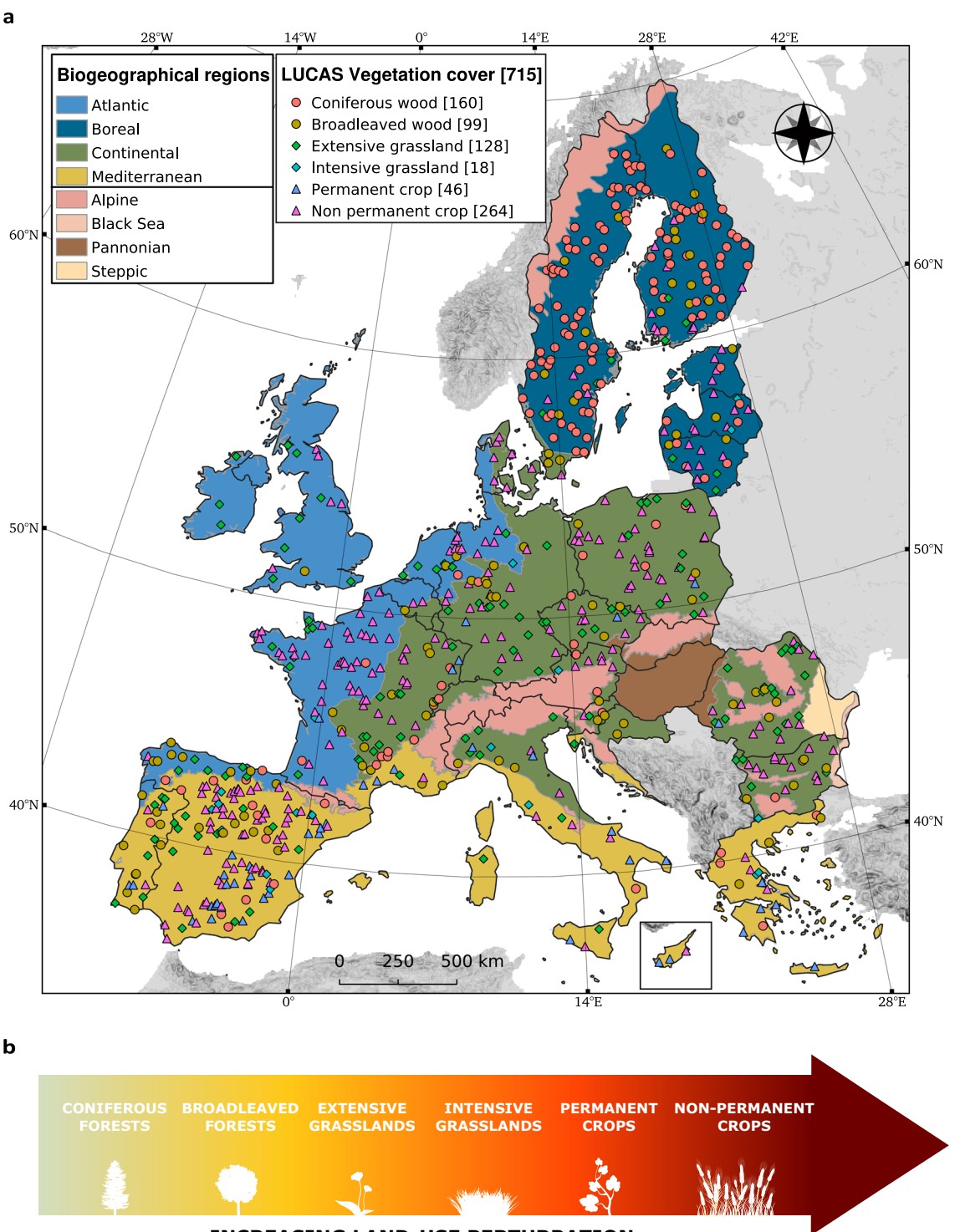

**Fig. 1 | Sampling design. a** Sampling points distribution coloured by vegetation cover type across biogeographical regions. The number of sites is indicated between brackets. **b** Vegetation cover types ordered along a gradient of increasing land-use perturbation.

(Supplementary Figs. 1, 2c). Similar investigations were conducted at the functional group-level and are presented in the Supplementary Figs. 4, 5 and in Supplementary Tables 3, 4.

**Single-effects of soil properties, climate, and vegetation cover**
When considering the impact of drivers as single (non-interactive) effects on microbial α-diversity, variation partitioning analysis showed that bacterial communities were mainly driven by soil properties (14.5% unique variance explained for the observed richness), followed

by climate (1.3%) and vegetation cover (0.6%). Conversely, fungal communities were shaped by vegetation cover (5.1%), soil properties (3.1%) and climate (0.6%) (Fig. 4a, b and Supplementary Fig. 6). Soil pH, calcium carbonate content and carbon-to-nitrogen ratio (C:N ratio) were the most important soil properties explaining bacterial α-diversity (Supplementary Fig. 7). Increasing pH values favoured both bacterial richness and Shannon index, while an increasing soil content in calcium carbonate and higher C:N ratio values showed the opposite effect. For fungi, once the main impact of vegetation cover was

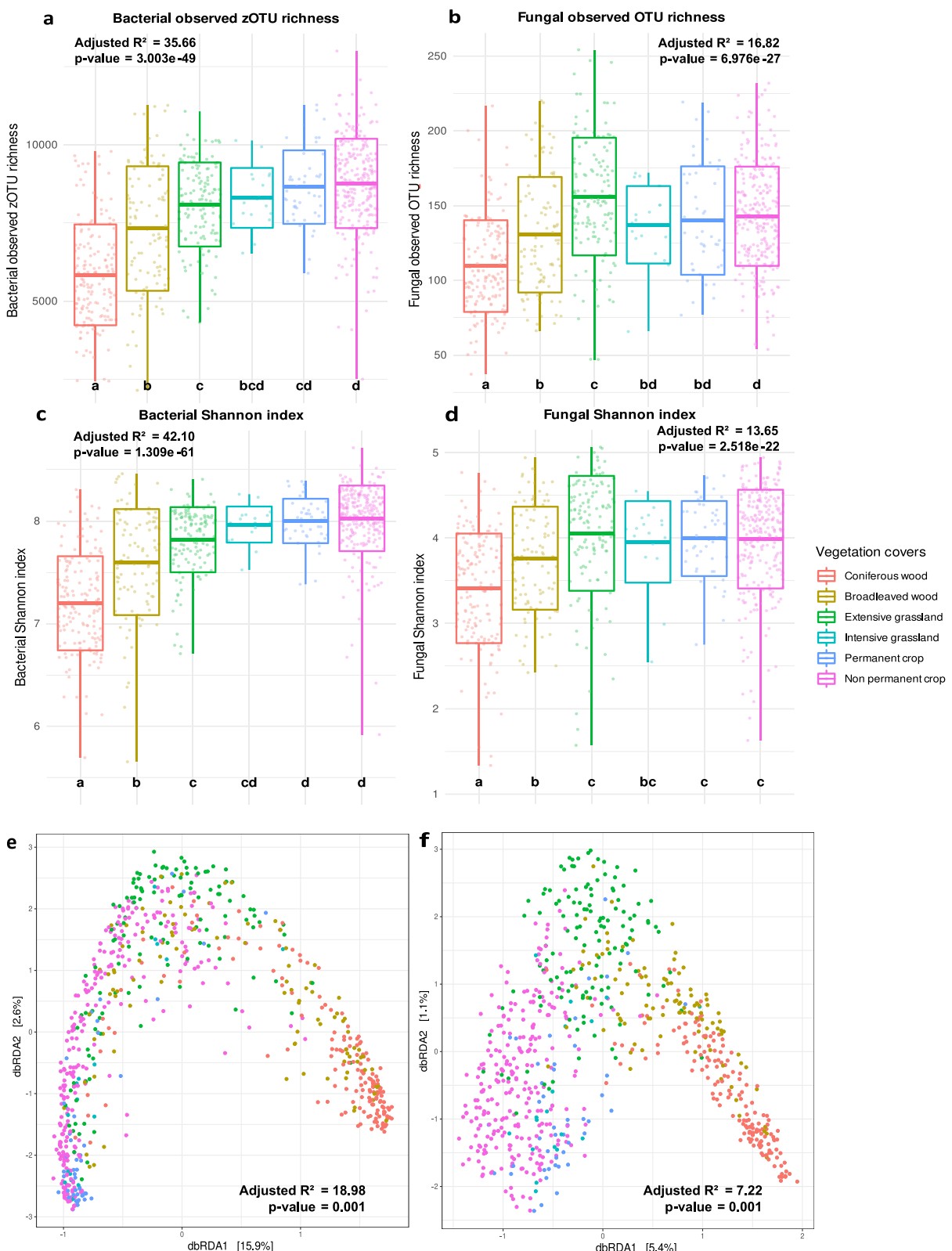

accounted for, silt content, bulk density and pH were the most influencing soil properties on fungal richness (silt content had a negative impact while bulk density and pH had a positive effect). Soil pH and extractable potassium had a positive and negative effect, respectively, on fungal diversity (Supplementary Fig. 7). The number of soil properties and climatic variables acting on bacterial richness and diversity was higher than that of fungal α-diversity metrics. Soil properties and

climatic variables impacting both bacterial and fungal α-diversity affected both groups in the same way (either positively or negatively), with the exception of temperature seasonality and silt content (Supplementary Fig. 7).

Considering β-diversity, soil properties had the strongest impact on bacterial (11.9% of unique variance explained) and fungal (2.5%) community composition (Fig. 4a, b and Supplementary Fig. 6e).

**Fig. 2 | Land-use perturbation (associated to vegetation cover types) and communities. a** Bacterial observed richness among vegetation cover types. **b** Fungal observed richness among vegetation cover types. **c** Bacterial Shannon index among vegetation cover types. **d** Fungal Shannon index among vegetation cover types. Data are presented as mean values ± standard deviation SD. Boxplot centre line indicates the mean, lower and upper hinges the standard deviation around the mean and each whisker correspond to the minimum and maximum values, respectively. Different letters correspond to a significant difference among proportions of (z)OTUs in compared vegetation cover types, and $p$ value corresponds to the one obtained with a Kruskal-Wallis test testing the vegetation cover effect. **e** Bacterial dbRDA plot testing the effect of vegetation cover on bacterial

community structure (β-diversity), based on the Bray-Curtis dissimilarity matrices calculated on the Hellinger-transformed sample-by-zOTU table. **f** Fungal dbRDA plot testing the effect of vegetation cover on fungal community structure, based on the Bray-Curtis dissimilarity matrices calculated on the Hellinger-transformed sample-by-OTU table. $P$ value corresponds to the one obtained from the one-way ANOVA test. In all panels, adjusted R-squares are expressed in percentages and colours represent the different vegetation cover types, where $n = 715$ total sites, with 160 belonging to coniferous woods, 99 to broadleaved woods, 128 to extensive grasslands, 18 to intensive grasslands, 46 to permanent crops and 264 to non-permanent crops sites. Source data are provided as a Source Data file.

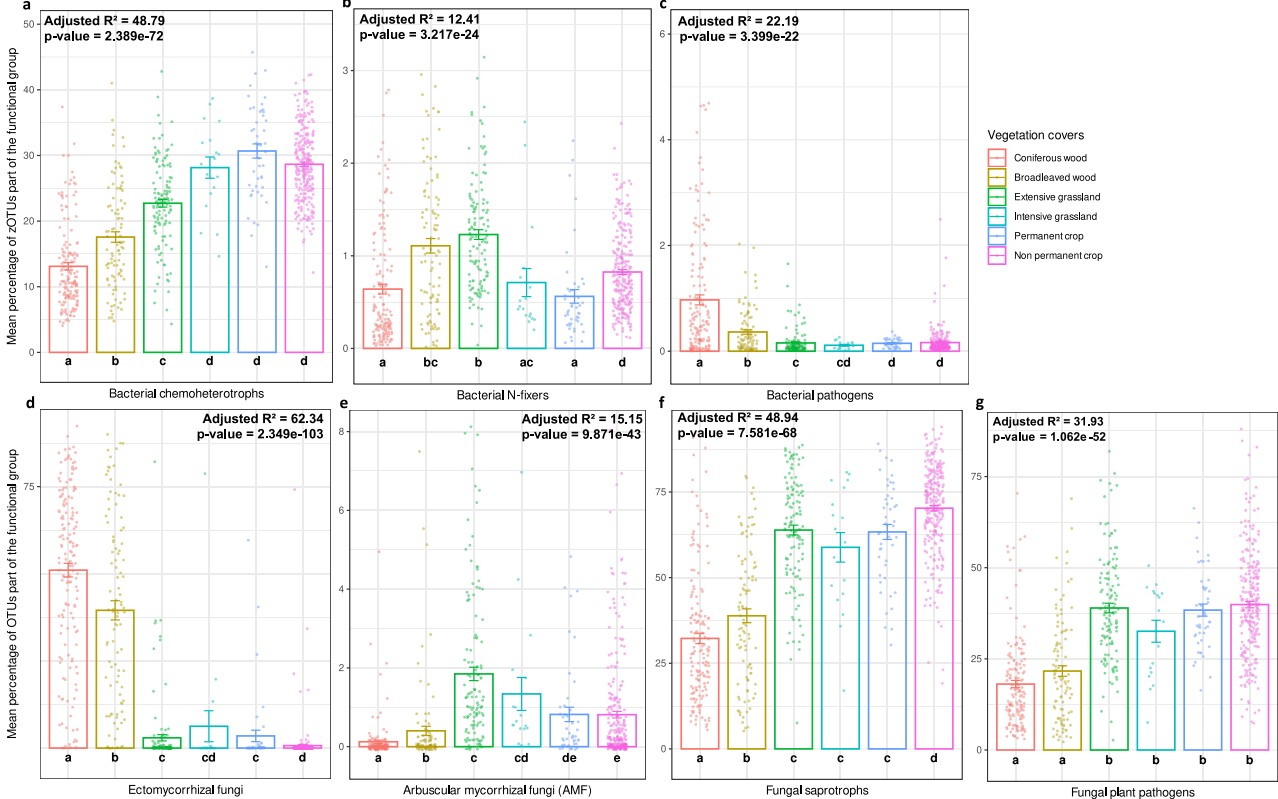

**Fig. 3 | Land-use perturbation (associated to vegetation cover types) and functional groups.** Mean relative proportion in percentages (±SE) of bacterial zOTUs and fungal OTUs (weighted by their read counts) belonging to a functional group, for each vegetation cover, detailed by functional group for (**a**) bacterial chemohetrotrophs, (**b**) bacterial N-fixers, (**c**) bacterial pathogens, (**d**) ectomycorrhizal fungi, (**e**) arbuscular mycorrhizal fungi (AMF), (**f**) fungal saprotrophs, (**g**) fungal plant pathogens. Error bars represent standard error. Different letters correspond to a significant difference between proportions of (z)OTUs in compared

vegetation cover types, and $p$ value corresponds to the one obtained with a Kruskal-Wallis test testing the vegetation cover effect. Adjusted R-squares are expressed in percentages, colours represent the different vegetation cover types, and $n = 715$ total sites, with 160 belonging to coniferous woods, 99 to broadleaved woods, 128 to extensive grasslands, 18 to intensive grasslands, 46 to permanent crops and 264 to non-permanent crops sites. Source data are provided as a Source Data file.

Climate was the second driver explaining a difference in bacterial community composition between sites (3.1%), while vegetation cover was the second driver explaining this difference for fungal communities (2.1%) (Fig. 4a, b and Supplementary Fig. 6f). Soil pH was the most important soil property explaining both bacterial and fungal β-diversity, followed by C:N ratio (Supplementary Fig. 8).

We also performed variation partitioning analyses to assess which variables best explain microbial functional groups, focusing on single factors (e.g. soil properties, climate, vegetation). We observed that the distribution of soil bacterial functional groups was mainly driven by soil properties, explaining 7.7%, 8.0% and 20.1% of unique variance explained for bacterial chemoheterotrophs, N-fixers and pathogens respectively (Supplementary Fig. 9a–c). Fungal functional groups were mainly shaped by

vegetation cover, representing 28.2% of unique variance explained for ectomycorrhizal fungi, 17.0% for AMF, 11.9% for saprotrophs and 12.0% for plant pathogens (Supplementary Fig. 9d–g).

Considering the soil conditions driving bacterial functional groups, bacterial chemoheterotrophs were associated to soils with high pH and low C:N ratio values (Fig. 4c). They were also favoured by more compacted soils (i.e. higher bulk density values) with higher potassium contents and low calcium carbonate and silt contents. N-fixing bacteria were more abundant in soils with low pH values as well as low phosphorus contents and C:N ratio values, low calcium carbonate and silt contents, but higher bulk density values (Fig. 4d). Potential bacterial pathogens predominated in more sandy soils (low clay and silt contents) with low pH values but higher calcium carbonate contents and bulk density values (Fig. 4e).

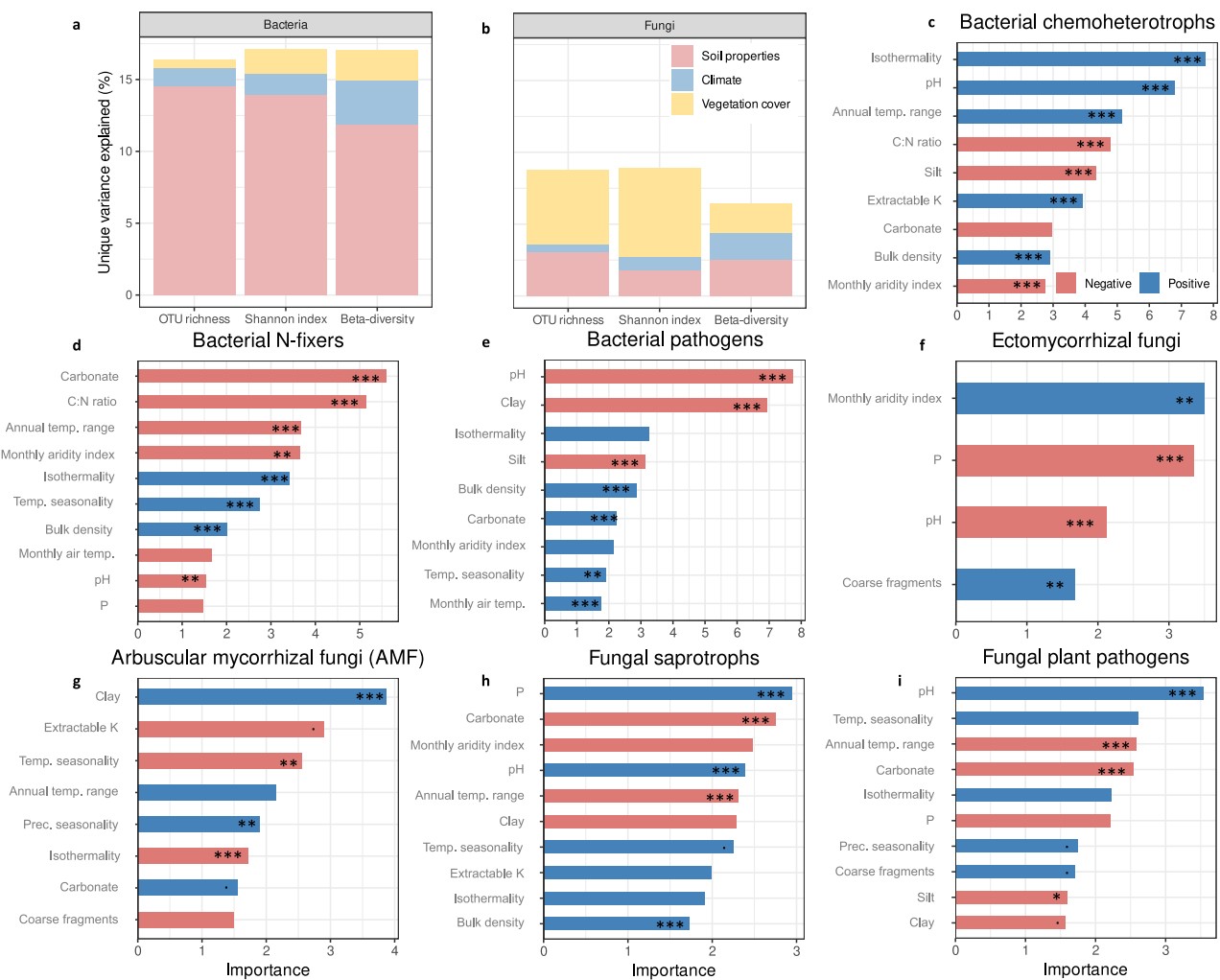

**Fig. 4 | Single effects. a** Part of unique variance explained by pre-selected soil properties and climatic variables and by vegetation cover on the bacterial observed richness, Shannon index and β-diversity. **b** Part of unique variance explained by pre-selected soil properties and climatic variables and by vegetation cover on the fungal observed richness, Shannon index and β-diversity. Percentages correspond to the unique partitioned variance of each single effect found after a variation partitioning testing for the effect of pre-selected soil properties, climatic variables, and vegetation cover (Supplementary Fig. 6). Colours correspond to the type of environmental variables (soil properties, climate or vegetation cover). **c–i** Bacterial or fungal variable importance for each soil property and climatic variable in explaining the proportion of (z)OTUs (weighted by their read counts) belonging to

a given functional group for (**c**) bacterial chemohetrotrophs, (**d**) bacterial N-fixers, (**e**) bacterial pathogens, (**f**) ectomycorrhizal fungi, (**g**) arbuscular mycorrhizal fungi (AMF), (**h**) fungal saprotrophs, (**i**) fungal plant pathogens Only the numerical properties selected by the models are presented, but the vegetation cover was selected as a significant important (categorical) variable as well and included in the models. Colours represent the positive or negative sign of the variable in the linear model found after feature selection. The stars represent the level of the p value for each term in the one-way ANOVA test (*** $p < 0.001$; ** $p < 0.01$; * $p < 0.05$; . $p < 0.1$). Exact p values are provided in Supplementary Data file 2. Source data are provided as a Source Data file.

Considering climatic variables (Fig. 4c–i), more humid conditions (i.e. higher values of the monthly aridity index) negatively impacted the proportion of bacterial chemoheterotrophs, while more contrasted temperature conditions over the year favoured them, e.g. higher isothermality (i.e. larger day-to-night temperature oscillations relative to summer-to-winter oscillations) and larger annual temperature range values. Bacterial N-fixers presence was favoured by higher isothermality values and higher variations in temperature seasonality over the year, but within a more restricted range of annual temperature (i.e. lower annual temperature range values), and by colder and drier climatic conditions (i.e. lower monthly air temperature and monthly aridity index values respectively). Contrasted and warmer temperature as well as more humid conditions favoured bacterial pathogens (i.e. increasing isothermality, monthly air temperature and monthly aridity index, and larger variations in temperature seasonality).

While the effects of vegetation cover on fungal functional groups was as seen in Fig. 3, taking into account soil properties, we found that ectomycorrhizal fungi were associated to soils with low pH values and low phosphorus contents, and high percentages of coarse fragments (Fig. 4f). AMF were relatively more abundant in soils with low potassium contents and percentages of coarse fragments, and high calcium carbonate and clay contents (Fig. 4g). Fungal saprotrophs were favoured by soils with high bulk density and pH values, high potassium and phosphorus contents, and low calcium carbonate and clay contents (Fig. 4h). Potential fungal plant pathogens were associated with sandy soils with high pH values, higher percentages of coarse fragments, and low phosphorus and calcium carbonate contents (Fig. 4i). More humid conditions favoured ectomycorrhizal fungi while hampering the presence of fungal saprotrophs. Wider ranges of annual temperatures promoted plant symbionts (e.g. AMF) over saprotrophs and pathogens. At the opposite, larger variations in temperature

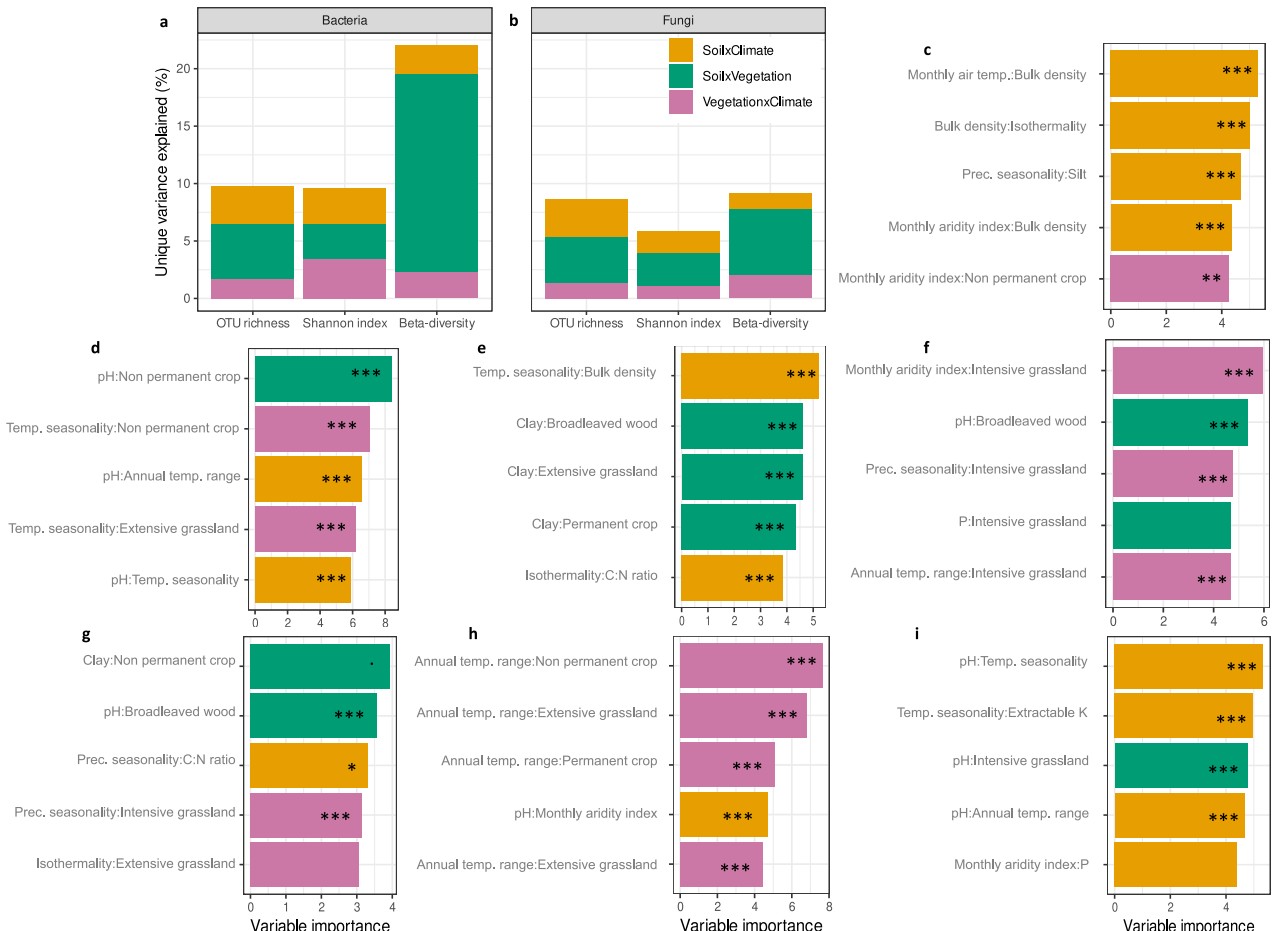

**Fig. 5 | Interaction effects. a** Part of unique variance explained by pre-selected interactions between soil properties, climatic variables and vegetation cover on bacterial observed richness, Shannon index and β-diversity. **b** Part of unique variance explained by pre-selected interactions between soil properties, climatic variables and vegetation cover on fungal observed richness, Shannon index and β-diversity. Percentages correspond to the unique partitioned variance of each interaction type found after a variation partitioning testing for the effect of pre-selected interaction terms between soil properties, climatic variables and vegetation cover (Supplementary Fig. 11). Colours correspond to the type of interactions (soil properties × vegetation, soil properties × climate or vegetation × climate). **c–i** Variable importance of the first five two-way interaction terms in the interaction models for bacterial and fungal functional groups for (**c**) bacterial chemohetero-trophs, (**d**) bacterial N-fixers, (**e**) bacterial (human) pathogens, (**f**) ectomycorrhizal fungi, (**g**) arbuscular mycorrhizal fungi, (**h**) fungal saprotrophs and (**i**) fungal plant pathogens. Bars are coloured by interaction type. The stars represent the level of significance of the $p$ value for each term in the one-way ANOVA test (*** $p < 0.001$; ** $p < 0.01$; * $p < 0.05$; . $p < 0.1$). Exact $p$ values are provided in Supplementary Data file 2. Source data are provided as a Source Data file.

seasonality favoured these last two groups over the AMF communities. Finally, precipitation only positively affected the proportion of pathogens and AMF.

### Interactive effects of soil properties, climate, and vegetation cover

In a next step, we used an ordination method and associated biplots to illustrate how community diversity and structure (α- and β-diversity) and the presence of potential functional groups were preferentially related to certain combinations of vegetation cover type, soil properties and climatic variables (Supplementary Fig. 10). For instance, croplands and grasslands with high pH values and clay contents were more frequent in arid and warm conditions (i.e. low monthly aridity index values and high monthly air temperature respectively) and hosted more bacterial chemoheterotrophs, while woodlands were characterised by high C:N ratio values, featured less-compacted soils (i.e. low bulk density values) and hosted more ectomycorrhizal fungi (Supplementary Fig. 10).

Variation partitioning showed that the most relevant interactions shaping bacterial and fungal richness, diversity and community composition were the ones involving vegetation cover. In particular, interactions among soil properties and vegetation cover (soil properties × vegetation) represented 4.8%, 4.0% and 2.8% of unique variance explained for bacterial and fungal observed richness and fungal diversity respectively (Fig. 5a, b and Supplementary Fig. 11). Those interactions (soil properties × vegetation) mainly explained the differences in community composition between sites (β-diversity) for both bacteria (17.3% of unique variance explained) and fungi (5.8%) (Fig. 5a, b and Supplementary Fig. 11). Bacterial diversity was relatively more affected by interactions among vegetation cover and climate (vegetation × climate), representing 3.4% of unique variance explained (Fig. 5a and Supplementary Fig. 11). More details about the most influential interactions are available in Supplementary Figs. 12, 13. Similar investigations were conducted for the functional groups and are presented in Fig. 5 and Supplementary Figs. 14, 15.

Overall, interaction models performed better than the single-effect ones. For both groups of microorganisms (bacteria and fungi) and inferred functional groups, interaction models explained from 2% up to 16% of additional variance for all considered metrics (α-diversity, β-diversity) and displayed smaller Akaike Information

Criterion (AIC) values (i.e. more parsimonious models) (Supplementary Table 5). However, for fungal Shannon index, AIC values were equivalent among model types (Supplementary Table 5). Interaction models also highlighted how certain soil properties differently affected communities and potential functional groups depending on the climatic and vegetation contexts. For instance, the interaction between pH and temperature seasonality was identified as one of the most influencing interactions on fungal communities and functional groups (Fig. 5 and Supplementary Fig. 12). A concrete visualisation of such interaction is given in supplementary Fig. 16, where cluster areas were defined based on temperature seasonality values. Increase in fungal richness, diversity and proportion of potential fungal plant pathogens with soil pH differed among climatic cluster areas (Supplementary Fig. 16a). Such increase was more pronounced in areas characterised by a high temperature seasonality but also differed among vegetation cover types (Supplementary Fig. 16b). This example highlighted the interaction effect among the three types of considered factors.

## Discussion

### Effects of land-use perturbation on microbial communities
Here we compared several microbial community metrics (i.e. richness, diversity, structure) among six major vegetation cover types in Europe. Land-use perturbation (i.e. established as a gradient from woodlands to highly-managed croplands) affected soil microbial diversity. Microbial richness (bacterial zOTUs and fungal OTUs) and diversity (Shannon index) increased from less disturbed (i.e. woodlands) to more managed areas (i.e. grasslands and croplands). In agreement with this, community structure also differed among vegetation cover types, with the communities in woodlands and croplands being the most dissimilar. In addition, shifts in relative abundances of main microbial phyla and classes were observed among croplands, grasslands and woodlands. These results confirm previous observations at a local scale showing that land-use intensification alters soil microbial community structure[32], and suggest that highly-managed habitats (e.g. croplands) represent a distinct soil biodiversity pool from less-disturbed sites such as natural or semi-natural systems (e.g. woodlands). Higher microbial richness and diversity in croplands and grassland compared to woodlands might result from increased niche availability at the local scale due to either soil perturbation or the presence of a heterogeneous environment with different plant species (e.g. in grasslands or when different crops are rotated in arable and vegetable fields). In contrast, woodlands might be a highly competitive environment with reduced niche opportunities where fewer microbial taxa can persist[33,34]. Interestingly, the observed below-ground patterns of microbial richness and diversity are not the same as for above-ground organisms, which often have higher richness and diversity in woodlands, and suggest contrasting responses of both diversity types to land-use perturbation[35].

### Effects of land-use perturbation on functional potential
We also showed that, other than taxonomic diversity, the functional annotation should be taken into account when addressing the impact of land-use perturbation on soil microorganisms, as microbial functional groups significantly varied from less disturbed to highly-managed areas. Here we inferred functional groups based on bacterial and fungal taxonomy and showed that land-use perturbation might promote antagonists, including potential fungal pathogens. The reduction of plant species richness in more managed areas may indeed allow fungal pathogens to occupy additional ecological niches due to homogenisation of the above-ground plant-host community, reducing dilution effects and allowing pathogens to spread faster and accumulate[35]. Our results also showed that broadleaved woods and extensively-managed grasslands hosted higher proportions of N-fixing bacteria. In contrast to croplands, usually fertilised with nitrogen, most woodlands and grasslands are not fertilised, and nitrogen is often limited in these environments. As such, N-fixing bacteria, as free-living N-fixers (e.g. *Telmatospirillum siberiense*) or in association with N-fixing plants (e.g. *Mesorhizobium loti*), may represent the principal source of nitrogen inputs for plants in such environments[36]. In addition, arbuscular mycorrhizal fungi (AMF) dominated in extensively-managed grasslands compared to intensively-managed grasslands and croplands, in line with previous observations at smaller scales showing a marked decrease in AMF upon land-use intensification[37,38]. Ectomycorrhizal fungi (EcM), known to be involved in symbiotic relationships with a broad diversity of forest trees[39,40], dominated woodlands. However, EcM fungi rarely associate with trees present in croplands and grasslands (e.g. orchards and sparse shrubs), and were therefore less abundant in these environments, suggesting a negative impact of land-use intensification on ectomycorrhizal symbiosis[41].

### Effects of land-use perturbation on biodiversity-ecosystem functioning relationship
A range of experimental studies demonstrated that microbial diversity promotes ecosystem functioning[4,42] and high microbial richness is generally viewed as positive for ecosystem functioning and the health of a wide range of organisms[43]. However, our study indicates that greater microbial taxonomic richness/diversity does not necessarily imply beneficial outcomes, as highly-perturbated soils, hosting higher taxonomic richness, harboured a greater prevalence of potentially undesired taxa (e.g. pathogens). Altogether, these findings confirm that an accurate evaluation of both taxonomic and functional biodiversity is necessary for an in-depth understanding of the impacts of land-use perturbation on the structure and functions of soil microbial communities at continental-scale. We argue that focusing on taxonomic diversity only might lead to a false assumption that increased biodiversity implies benefits in terms of ecosystem functioning. In particular, we highlight that, besides the taxonomic monitoring of soil microorganisms, a scheme for controlling functional groups could help conceiving actions targeting soil management and preservation. In principle, the systematic assessment of potential microbial functional groups would not require any additional costs, as it relies on databases (e.g. FAPROTAX, FungalTraits) predicting functions based on inferred taxonomy. This approach would allow a first fast-screening of potential microbial functions, although it has some limitations further discussed below.

### Single drivers of microbial communities
We also investigated the forces driving diversity and structure of fungal and bacterial communities at large scale, by considering the drivers acting separately. Interestingly, climate-related variables were not the main driver of the assembly of soil microorganisms in this study. Soil properties were important predictors of bacterial diversity, richness and composition. Apart from the well-documented effects of pH and C:N ratio[10,11,24,44–47], other parameters, such as potassium, calcium carbonate, and the soil content in silt and clay, which are often overlooked, played a significant role in shaping bacterial communities. These findings illustrate the need to expand the range of variables to be examined when investigating soil bacterial distribution. On a more practical level, they demonstrate why specific actions on soil management (i.e. practices affecting soil properties) should be considered when targeting soil bacterial communities and related functions and ecosystem services for conservation purposes.

The fungal α-diversity was driven by the vegetation covering the soil, more than climate and soil properties as previously reported[48]. This may be due to the tight link between fungi and plants (e.g. endophytism, mutualism−e.g. mycorrhizal fungi−or plant pathogenicity[49–51]). Fungal β-diversity was, however, mainly driven by soil properties. In particular, differences in community composition were found to be highly influenced by soil pH. The driving effect of pH on fungal β-diversity has already been reported but few reports are available so far

at such a large scale and these studies are limited to a few vegetation cover types (e.g. grasslands and woodlands[11,24,26,52,53]). From a management perspective, our results suggest that when targeting soil fungi, actions on vegetation cover (e.g. rewilding) combined with soil management could help ensure more effective conservation measures.

## Single drivers of functional potential

Compared to taxonomic diversity, the factors driving soil microbial functions received less attention. However, a better understanding of how taxonomic diversity patterns can be associated to ecological functions is key to ensure the provision of ecosystem services that rely on such functions[54]. Recent studies have focused on the factors driving functional diversity within soil fauna[55], but large-scale studies on the factors driving microbial functional diversity are still limited[26,48], particularly for bacteria. To fill in those knowledge gaps, we used taxonomy to predict functional groups within the bacterial and fungal communities and tested single-effect models on them. We observed that microbial functional groups were not driven by the same factors among groups. In particular, functional groups were influenced by different sets of soil properties and climatic variables, suggesting that conservation actions could have different impacts on various functional groups.

Our findings also offered new insights in relation to soil properties and climate. As of soil features, for instance, bacterial chemoheterotrophs were preferentially found in highly-managed lands (i.e. croplands) and were positively affected by bulk density, a soil characteristic often overlooked when investigating the effects of soil properties on microbial communities[56]. Increased chemoheterotrophy could thus be related to agricultural practices and also indicate high quality litter inputs promoting mineral-associated organic matter formation[57,58]. Similarly, fungal saprotrophs and fungal plant pathogens were negatively linked to calcium carbonate content while AMF were positively linked to it. Studies linking calcium carbonate to fungal functional groups at a large scale are virtually missing and further investigations could contribute to a better assessment of lime application impacts on fungal functional groups[59]. As the relationship between the addition of calcium carbonate and deacidification of soil pH is non-linear[60], being able to relate both lime application and targeted pH correction to variations in fungal symbionts and fungal pathogens in soils may help to quantitatively assess the impacts of this agricultural practice on soil functioning, and contribute to better decisions on soil management. Our findings also confirm the need to deepen the knowledge of drivers of soil microbial functional groups by including as many soil physicochemical properties as possible, next to information on the vegetation covering the soils (especially for biotrophic organisms) and climatic conditions.

Looking at climate, warmer conditions were found impacting beneficial functions for plant communities as, for example, a drier climate favoured bacterial chemoheterotrophs but reduced ectomycorrhizal symbionts presence. Both potential bacterial human pathogens and fungal plant pathogens were favoured by high temperature seasonality values, while those conditions hampered AMF presence. Interestingly, our results demonstrate the difficulty in assessing one overall effect (positive or negative) of climate on soil microbial functions, and highlight the need to investigate a large set of climatic variables to better capture the impact of temperature and precipitation evolution (within and over a year) on each functional group. Considering a potential future homogenisation of communities in an increasingly dry climate[61,62], our findings suggest that changes in climate may impact the provision of key ecosystem services by selecting for some specific taxa and associated functions. Simultaneously, climatic extremes (e.g. increased temperature and drought) may promote plant vulnerability to pathogens[63]. Using climatic projections may help assessing the future impacts of climatic change on functional groups and, thus, designing preventive actions and monitoring schemes.

## Interactive drivers of microbial communities and functional potential

Our study also showed that interactions among soil properties, climate, and vegetation cover better explain patterns of soil microbial communities and inferred functional groups than when considering drivers acting separately. Models accounting for these interactions did not only perform better (i.e. from 2% up to 16% of additional explained variance and, overall, equivalently or more parsimonious models) but also brought additional information on potential drivers of soil microbiomes in different climatic and vegetation contexts. For instance, single-effect analyses hid the impact of climate by showing a clear predominance of soil properties in explaining bacterial community patterns, while interaction analyses highlighted that variations in temperatures (annual range and seasonality) are particularly influential on bacterial diversity of croplands- and extensively-managed grasslands-associated communities, in combination with soil variables. In addition, interaction analyses confirmed the dominance of vegetation cover in explaining fungal richness and diversity, and also underlined the importance of certain soil properties (e.g. clay content) that were not identified as relevant drivers in the single-effect analyses. Interestingly, interaction-effect models on β-diversity detected a limited role of climate for both bacteria and fungi, indicating that the diversity of communities between sites at broader scales is preferentially driven by soil properties interacting with vegetation cover. This might reflect the fact that at the site scale, the uniqueness of communities is promoted by local and more stable conditions (i.e. soil properties and vegetation cover), more than by long-term acting factors (i.e. climate)[64].

Furthermore, our results encourage the identification of climatic cluster areas where the impact of soil properties in relation to different vegetation cover types should be further studied. To investigate soil microbial communities, large areas may be broken down into combinatorial patches of environmental features (e.g. climate, vegetation and soil properties) known to affect microorganisms through their interactions. Such procedure may facilitate the establishment of clusters of action (areas of priority) where implementing appropriate monitoring tools and preservation measures. This might pave the way to a new strategy overcoming the eternal difficulty, faced by the scientific community, to propose reliable approaches for soil (microbial) diversity conservation[65].

## Study limitations and perspectives

Despite the broad range of factors included in our analysis, the applied models still recorded a significant unexplained variance, suggesting other forces playing on soil microbial communities. Previous studies have identified variables as potential drivers of soil microbial diversity (e.g. soil wilting point[66]) and community structure (e.g. specific micronutrients[67]) that LUCAS survey does not currently cover due to the difficulty of collecting related field data over such a large area. Also, future studies should compare multiple sampling points within the same location, in order to account for micro-scale variability of the biota (i.e. changes over distances of few metres). Additionally, we noted that the lack of quantified information on covering plants (e.g. plant community richness and structure, crop yield) may have represented a limitation to our interpretations (e.g. when comparing seminatural and highly-perturbated soils and when approaching the distribution patterns of biotrophs) and could also contribute to the unexplained variance[15].

Furthermore, the proportion of soil taxa that were assigned to a potential function remained low, especially for bacteria. Thus, efforts to improve the taxonomic and functional characterisation of microbial communities and specific microbial taxa would be needed to provide further insight into the interactive effects among soil properties, vegetation cover, and climate on soil microbial functional groups. In particular, such efforts should be made in all types of vegetation cover,

to thoroughly address potential biases in inferred functional annotation among more anthropic and semi-natural environments. This is especially true for potentially pathogenic taxa, that are more frequently studied in croplands areas so far[68]. Also, a range of potentially pathogenic fungal taxa are known to have saprotrophic activities; it is likely that the actual proportion of pathogenic taxa is lower and the reported proportion of potential fungal pathogens (e.g. almost 40% for grasslands and croplands) represents an upper limit and includes taxa that act as saprotrophs but are also classified as opportunistic pathogens.

Although functional databases offer a fast-functional screening of microbial data, we were aware of the limitations regarding their functional assignments[69]. For instance, bacterial zOTU assignments were done based on 16S, a conserved DNA region that can discriminate bacteria at the finest taxonomic level (i.e. strain), while it is likely that some functions may not be phylogenetically conserved (i.e. different strains of same species carrying out different functions)[70]. An analysis based on inferred functions, thus, would benefit from additional experimental works like (i) metagenomics, metatranscriptomics and metaproteomics to quality-check the predicted functional annotation (metagenomic) and assess the functionally active communities (metatranscriptomics and metaproteomics[71,72]), (ii) metabolomics to better understand community functional potential by quantifying the presence of functional products (i.e. metabolites) in the environment[73], and (iii) a cause-effect analysis (e.g. linking the occurrence of putative plant pathogens to detrimental plant growth and reduced yield in agricultural fields). These approaches, if applied to LUCAS Soil, could steer the functional assessment of soil microbial communities across Europe.

## Guidelines for policy-related applications

Despite the above-mentioned limitations, our findings provide guidelines for the establishment of priorities and targets, which are currently missing, for soil biodiversity monitoring within the European Union (EU). The EU, like most countries, lacks specific legislation for soil biodiversity protection. This gap is mainly due to the lack of robust approaches for (i) identifying drivers upon which preservation and monitoring actions could be designed and (ii) defining spatial areas where those actions should apply. Nonetheless, policy initiatives taken recently at the EU level (e.g. EU Biodiversity Strategy for 2030, Farm to Fork Strategy and EU Soil Strategy for 2030 with a proposal for a Soil Health Law[74–76]) show a fertile ground for the possible development of a legal framework that takes into account soil life[77]. Our analyses may thus lay the foundations of a new paradigm for preserving soil microbial communities and promoting ecosystem services provided by soil functional groups, with two principles at its base:

1.  Investigating the interactions of a large set of soil properties and climatic variables (including poorly considered parameters), as well as various vegetation cover types, helps explaining large scale patterns for soil bacteria and fungi, and their functional groups. The outcomes derived from interaction analyses can be used to circumscribe cluster areas featuring specific environmental properties in terms of soil, vegetation and climate. Preservation actions, monitoring schemes and thresholds may be tailored on these well-defined zones, and adapted to the organism type or functional group.
2.  Decisions based on the functional diversity should complement the taxonomical ones for preservation and monitoring purposes, by keeping in mind that a high soil microbial (taxonomical) diversity might not always be beneficial (higher potential pathogenicity). The functional groups to be prioritised (likely organisms providing beneficial functions, at least from an anthropogenic point of view) have thus to be decided and then better monitored to control the effectiveness of measures.

In conclusion, this study demonstrates that microbial diversity, community structure and potential functional group distribution vary along a gradient of increasing land-use perturbation. At European scale, microbial communities from more disturbed areas (e.g. croplands) differ the most from communities associated to semi-natural habitats (e.g. woodlands). Soils harbouring richer and/or more diverse communities (i.e. croplands and grasslands) are also characterised by a higher presence of potential fungal pathogens. In contrast, woodlands and extensive grasslands harbour more fungal plant symbionts and N-fixing bacteria. Our study also investigates three types of environmental drivers (i.e. soil properties, climate and vegetation cover) and provides evidence that bacterial diversity is mainly shaped by soil conditions while fungal diversity is mainly influenced by vegetation cover. Differences in microbial community structure among sites are better explained by variations in soil properties. Our analyses show that the distributions of inferred microbial functional groups are shaped by different type of drivers (i.e. soil properties for bacteria but vegetation cover for fungi) and that the same soil and climatic conditions can have opposite effect on groups.

Overall, our results highlight that monitoring and preservation schemes should take both taxonomic and functional diversity into account in order to get a reliable estimate of the impacts of land-use intensification and environmental factors on below-ground diversity. While considering environmental drivers as acting in parallel (i.e. single effects) gives an adequate overview of microbial patterns at the European scale (up to 65% of variance explained), considering their interactions may lead to new perspectives with respect to soil diversity monitoring and protection.

## Methods
### Sampling and soil properties analysis
As part of the 2018 LUCAS Soil module, 881 sites were sampled across all European Union countries (EU) and the United Kingdom (the UK was an EU Member State at the time of the sampling). In brief, sampling points were selected based on a simulated annealing sampling applied to the LUCAS Soil 2009 data as the original population from which to sample. Point selection was performed so that an optimal configuration was found replicating a wide range of environmental variables (i.e. soil physico-chemical properties, topography, climate and land cover). At each location, five subsamples covering a depth of 20 cm were collected and mixed together. One subsample was collected at the precise geographical location of the pre-selected point while four additional subsamples were collected at the four cardinal directions (North, East, South and West), at a distance of 2 m from the first subsample location in each direction. Altogether, 500 grams of soil were stored on ice and transported to the JRC within 48 h of collection where they were then frozen. On completion of the survey, samples were sent to the molecular biology laboratory of the Mycology and Microbiology Centre (University of Tartu) for DNA analyses. A second sample was analysed for physical and chemical soil properties by SGS Hungary. Further details about LUCAS Soil sampling campaign are available in Orgiazzi et al.[29].

### Soil DNA analysis
Soil samples were analysed for their bacterial and fungal biodiversity through a DNA metabarcoding approach. Primer sets for barcode amplification were 515F (GTGYCAGCMGCCGCGGTAA) and 926R (GGCCGYCAATTYMTTTRAGTTT) for the bacterial 16S region[78,79] and ITS9mun (GTACACACCGCCCGTCG) and ITS4ngsUni (CGCCTSCSCTT ANTDATATGC) for the fungal ITS region[80,81]. Sequencing was performed by Illumina MiSeq platform with 2 × 300 paired-end mode for bacterial data and PacBio Sequel II platform for fungal data. Protocols for DNA extraction and amplification are described hereafter.

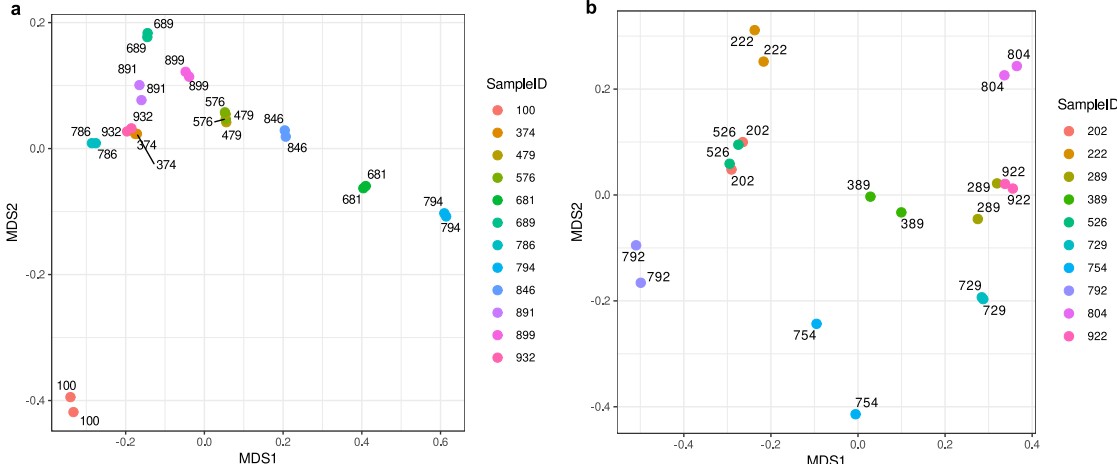

**Fig. 6 | Reproducibility of sequencing results.** Ordination plots for pairs of resequenced samples (replicates) for (**a**) bacteria and (**b**) fungi. Colours correspond to replicates with the same sample ID. Source data are provided as a Source Data file.

**DNA extraction and amplification.** DNA was extracted with the Qiagen DNeasy PowerSoil HTP 96 Kit Q12955-4. Three 0.2 g aliquots per sample were extracted. The three subsamples were pooled after extraction. A negative control and positive control were used during extraction to locate any external contamination and cross-contamination. Quality check and quantification of DNA were performed with Qubit™ 1X dsDNA HS Assay Kit using Qubit 3 fluorometer (Invitrogen).

All PCRs were performed in three replicates using 5 × HOT FIRE-Pol® Blend Master Mix (Solis BioDyne, Tartu, Estonia) in 25 µl volume. The optimal number of cycles and optimal annealing temperature were used for the primer pairs. In case of PCR failure, the extracted DNA was purified using Favorgen FavorPrep Genomic DNA Clean-up Kit FAGDC001-1 and the PCR was repeated with more cycles to the thermocycler programme if needed. For bacteria, the primers 515F and 926R were used. The PCR conditions included: 55 °C annealing temperature, 26 cycles, 1.5 ng/µl of DNA template (1 µl). In case of PCR failure even after DNA was purified, 28 cycles were used instead of 26 cycles. For eukaryote ITS region, the primers ITS9mun and ITS4ngsUni were used. The PCR conditions included annealing temperature 55 °C, 30 cycles, 1.5 ng/µl of DNA template (1 µl). In case of PCR failure even after DNA was purified, 33 or 35 cycles were used instead of 30 cycles. Both the forward and reverse primers were tagged with a 12-base multiplex identifier (MID) tag. The three replicates of each reaction were pooled and visualised on TBE 1% agarose gel. PCR products were purified using UltraClean 96 PCR Cleanup Kit (Qiagen). DNA concentrations were measured with Qubit™ 1X dsDNA HS Assay Kit using Qubit 3 fluorometer (Invitrogen).

**Metabarcoding library preparation.** Illumina amplicon libraries were generated using TruSeq DNA PCR-Free High Throughput Library Prep Kit with TruSeq DNA CD Indexes. For PacBio, SMRTbell library preparation followed precisely the Pacific Biosciences Amplicon library preparation protocol. Metabarcode sequencing was performed using the Illumina MiSeq platform with 2 × 300 paired-end mode or PacBio Sequel II platform. Positive and negative controls of extractions as well as amplifications were used to further infer any contamination and index-switching. For both bacterial and fungal DNA datasets, 12 randomly selected samples were replicated during library preparation to verify that library-to-library variability was low. Data were processed as described for the full dataset until normalisation of the (z)OTU table (threshold 16S: 12,277 counts and ITS: 458 counts). For fungi, two resequenced samples were excluded from the analysis, as one of the replicates in each pair had a much smaller number of reads than the

threshold. In unconstrained NMDS ordination based on Bray-Curtis dissimilarity, the pairs of replicates grouped by sample ID, confirming the reproducibility of sequencing results (Fig. 6). For fungi, some of the replicate samples may be more variable than others due to:

(i) differences in amplification efficiency, namely using longer amplicons (full-length ITS region) with the PacBio platform may result in different amplicon lengths, with shorter amplicons amplifying more efficiently than longer ones, so a small difference in DNA template concentration may be influential;

(ii) technical factors, namely the sequencing depth in PacBio is lower compared to the Illumina platform, hence the fungal dominance and the compositional nature of the sequencing data may result in some low abundance species being missing in a sample, resulting in different compositions of non-abundant species, and hence, in a higher dissimilarity.

Considering expected richness in samples, different sequencing strategies and sequencing depth were selected for bacteria and fungi. For bacteria, Illumina MiSeq was used and the samples with less than 50,000 reads were subjected to resequencing. For fungi, PacBio Sequel II instrument was used and the samples with less than 3000 reads were subjected to resequencing; for samples with less than 1500 reads new tags were selected and then resequenced. The sequencing was performed on 1050 samples of which 885 were official LUCAS Soil samples (from 881 locations). Samples from four locations were taken and, thus, sequenced twice due to sampling issues (DNA reads from 885 samples in total). Only DNA sequences from the four re-sampled soils were considered for further analyses (i.e. 881 unique samples). The bacterial amplicons (primers 515F and 926R) produced a total of 657,641,306 reads (140,032,087 accepted reads without control samples). The mean read count per sample was 133,364. 63 samples had less than 50,000 reads initially. Re-sequencing of those 63 samples provided 2,185,911 reads with the mean read count of 34,697 per sample. Out of 63 re-sequenced samples, 58 samples had less than 50,000 reads. The re-sequenced reads were combined with the original dataset. Altogether 9 samples had less than 50,000 reads in total and can be considered as substandard. A single sample had less than 5000 reads and can be considered as failed. The great loss of reads in the demultiplexing stage is attributable to the presence of large amounts of metagenomics material that varied greatly among samples. The fungal amplicons (primers ITS9mun and ITS4ngsUni) produced a total of 9,156,823 accepted reads (9,031,552 reads without control samples). The mean read count per sample was 8601. 46 samples had less than 3000 reads initially. Re-sequencing of those 46 along with 12 randomly selected samples produced 369,537 reads with the

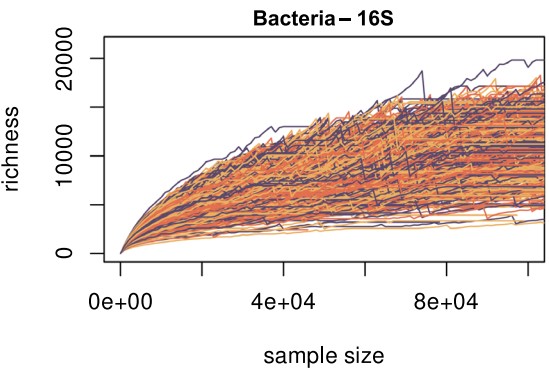

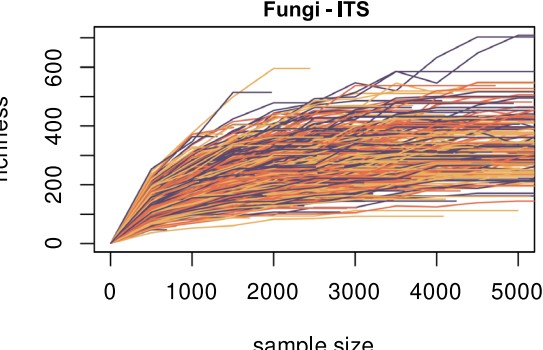

**Fig. 7 | Normalisation (SRS) curves for bacterial and fungal dataset (16S and ITS).** The curves are plotted up to 100,000 reads for bacteria and 5000 reads for fungi to better visualise small samples behaviour. The minimum threshold to which the sample-by-(z)OTU table is normalised: 40,109 reads for bacteria and 502 reads for fungi. Curves were generated using SRScurve function from SRS package[93]. Note: the zigzag behaviour of the SRS curves results from the combination of scaling and ranked subsampling of the fractional part (Cfrag) inherent to the SRS approach.

mean read count of 6371 per sample. The five samples with read count less than 1500 in the original run were assigned different tags and re-sequenced again. A total of 24,090 reads were produced with the mean read count of 4818 per sample. A single sample had a total read count of less than 500 and can be considered as failed. Each sample was evaluated based on read counts of original and both sets of re-sequenced reads, which were either replaced, merged, or retained in separate files.

**Sequence processing and filtering.** The Illumina and PacBio amplicon data (for bacteria and fungi, respectively) were demultiplexed using LotuS2[82] with default options. Paired-end reads were assembled using FLASH 1.2.10[83] with default options (minimum overlap 10 bp). Unmerged reads were removed from the final datasets.

For bacteria, zero-radius operational taxonomic units (zOTUs) were generated with UPARSE[84] (usearch version 10.0.024) according to the tutorial available for paired-end Illumina data (drive5.com/uparse/). Paired-end reads were merged with usearch. Merged reads were then truncated up to the 16S primer sequences (515F and 926R) and filtered for the presence of both primer sequences with a custom python script allowing up to 2 mismatches per primer. Primers were clipped. Merged reads were further quality-filtered with usearch by removing sequences with more than one expected error over the whole length (as suggested by Edgar & Flyvbjerg[85]). Duplicated sequences were collapsed with fqtrim[86] (version 0.9.7) and denoised with usearch. 136,427 zOTUs sequences were obtained and annotated with the taxonomy data available from the Ribosomal Database Project[87] (bacterial sequences, version 16) with usearch using a confidence-threshold of 0.8. Sequence filtering and trimming was done with fastp using default settings: reads that contain more than 40% bases with a PHRED quality score < 15 were removed. ZOTU abundances were finally obtained by counting the number of sequences matching to the zOTU sequences. To avoid sequencing artifacts, only the zOTU sequences with 50 counts at least across the samples and present in at least 10 samples were kept.

For fungi, data were processed as described in Tedersoo et al.[88] (see *Molecular analyses*), with the exception that singletons were removed from the analysis. 98%-OTUs were obtained and preferred over the use of ASVs, as ASVs are not considered suitable for full-length ITS sequences and are not optimal due to random PCR errors and the presence of multiple/highly similar copies of the ITS region in eukaryote genomes[89]. In addition, the use of ASVs increases the elimination of taxa that are both rare and phylogenetically unique[90]. Fungal taxonomic assignment was performed using BLAST + 2.11.0[91] by running MegaBLAST queries of representative

OTU sequences against the updated UNITE 9.1[92] beta reference dataset. These taxonomic assignments were checked against the 10 best megablast hits. Following taxon-specific thresholds were set: kingdom, $e_{max} = e^{-50}$; phylum, $e_{max} = e^{-55}$ to $e^{-80}$; class, $e_{max} = e^{-70}$ to $e^{-100}$; order, $e_{max} = e^{-80}$ to $e^{-120}$; genus, sequence similarity to the best match > 85–95%[88].

The quality-filtered reads were thus grouped into zero-radius operational taxonomic units (zOTUs, also known as ASVs, i.e. 100% similarity threshold) for bacteria and into OTUs at a 98% similarity threshold for fungi. Archaeal, chloroplasts and mitochondrial zOTUs were removed from the 16S dataset, which collectively accounted for 0.34% of all initial zOTUs. The sample-by-OTU tables resulting from those processes were then subjected to several filtering steps. Only sites belonging to biogeographical regions presenting all land cover types of interest (i.e. woodland, grassland and cropland) and a sufficient number of reads for the normalisation steps were kept for further analyses. The filtered datasets containing information of the 715 remaining sites were normalised to comparable sequencing depth (bacteria: 40,109 reads; fungi: 502 reads) using the Scaling with Ranked Subsampling (SRS) method[93].

In details, sites were first filtered by land cover type. To define land cover, we referred to LUCAS nomenclature, namely "the physical cover of the Earth's surface"[94]. Among the seven official LUCAS land cover classes, three main classes were selected to conduct robust statistical analysis: (i.e. cropland, grassland and woodland). Altogether 815 sites remained, representing over 90% of the total number of sampling locations. These 815 sites were distributed across the eight European biogeographical regions: Alpine, Atlantic, Black Sea, Boreal, Continental, Mediterranean, Pannonian and Steppic regions[95]. Only the regions harbouring sites belonging to all land cover types of interest were selected, namely the Atlantic, Boreal, Continental and Mediterranean regions, which resulted in inclusion of 752 sites. Finally, for fungal data, the sites with less than 500 read counts were excluded from further analysis in order to perform data normalisation (Fig. 7), leaving 715 sites for statistical analyses. In order to compare the bacterial and fungal datasets, the number of sites considered for bacteria was reduced accordingly.

Although the selected threshold of 502 read counts covers less initial richness than higher thresholds (Table 1), the normalisation of the sample-by-OTU table down to this sequencing depth led to similar results and conclusions compared with normalisation performed at higher sequencing depths (e.g. 1000, 1500 and 2000 read counts), both quantitatively (Table 2) and qualitatively. However, selecting higher thresholds discarded a non-negligible number of sites (Table 1), that removed valuable information (e.g. combination of vegetation cover and bioclimatic context) from the dataset and led to unbalanced

representation of the vegetation cover, soil and climatic conditions found across Europe. Normalisation of the data down to 502 read counts thus permitted us to (i) investigate a representative ensemble of sites in terms of vegetation cover, soil and climatic conditions across Europe, and (ii) to compare both bacterial and fungal communities at a large scale, from the same sampling sites (and thus, exact same environmental conditions). Values taken by the observed fungal OTU richness and Shannon index among different thresholds across common (i.e. non-discarded) sites are correlated with $r^2 > 0.92$ and available in Supplementary Data file 1.

**Functional annotation.** The functional traits databases FAPROTAX[96] and FungalTraits[97] were used to associate potential functions to bacterial zOTUs and fungal OTUs based on their taxonomy. Taxonomically unannotated bacterial zOTUs and fungal OTUs were not associated to any function and left as unknowns. For bacteria, FAPROTAX confidently associates a function to zOTUs identified at family, genus or species levels (i.e. a zOTU identified down to the order level is usually ignored by FAPROTAX, see the *Instructions* section on http://www.loucalab.com). For fungi, as described in Tedersoo et al.[7], functional annotation of OTUs was performed at the level of genera for most fungal guilds. Authors also used order-level annotation of certain life history traits (e.g. life form and arbuscular mycorrhizal fungi) when this was unequivocal for the entire order. Ectomycorrhizal fungi were additionally annotated at the level of sequence accessions based on information accumulated in UNITE.

For bacteria, the predicted functions were then grouped into broad functional groups including chemoheterotrophs (involved in the carbon cycle), nitrogen-fixers and human pathogens. For fungi, functional groups included ectomycorrhiza, arbuscular mycorrhiza, saprotrophs and plant pathogens. In details, bacterial zOTUs associated to the functions "chemoheterotrophy", "aerobic_chemoheterotrophy", "anaerobic_chemoheterotrophy" were grouped as potential bacterial chemoheterotrophs. Bacterial N-fixers

were associated to "nitrogen_fixation" and bacterial human pathogens to "human_pathogens_all", "human_pathogens_pneumonia". Fungal plant pathogens gathered OTUs identified as "root-associated" pathogens, "leaf/fruit/seed_pathogen", "other_plant_pathogen", "algal_parasite", "wood_pathogen", "root_pathogen", "leaf/fruit/seed_pathogen, algal_parasite", "wood_pathogen, leaf/fruit/seed_pathogen", "moss-associated" pathogens, "moss_parasite". Fungal saprotrophs gathered OTUs identified as "soil_saprotroph", "litter_saprotroph", "unspecified_saprotroph", "dung_saprotroph", "wood_saprotroph", "nectar/tap_saprotroph", "sooty_mold", "pollen_saprotroph". Arbuscular mycorrhizal fungi (AMF) were associated to "arbuscular_mycorrhizal" and ectomycorrhizal fungi to "ectomycorrhizal" as primary function, respectively. As several functions can be associated to each (z)OTU, some taxa could be identified as bacterial chemoheterotrophs and N-fixing bacteria, or as a potentially pathogenic chemoheterotrophic bacteria, but no overlap among N-fixing bacteria and pathogens was detected. Similarly, potential fungal plant pathogens were mainly identified as fungal saprotrophs as well.

Proportions of (z)OTUs belonging to the same functional group were estimated for each site, by weighting the (z)OTUs belonging to a given group by their number of read counts, and dividing the result by the total number of read counts of the site.

We also investigated potential major differences in functionally unassigned (z)OTUs among vegetation cover types and found that a total of 65,206 bacterial zOTUs and 7642 fungal OTUs were not assigned to any functional group at all (even outside previously mentioned groups of interest). This represents 81.9% of all bacterial zOTUs and 29.4% of all fungal OTUs. For each sampling site, two statistics were calculated: the proportion of (z)OTUs represented by the functionally unassigned (z)OTUs (number of functionally unassigned (z)OTUs for site A/total number of (z)OTUs detected in site A), and the proportion of read counts represented by the functionally unassigned (z)OTUs (sum of read counts belonging to functionally unassigned (z)OTUs for site A/total read counts for site A). We then calculated the mean and standard error of proportions by vegetation cover type and performed a Kruskal-Wallis test and a Pairwise Wilcoxon posthoc test (with a Benjamini-Hochberg's correction) to assess significant differences in proportions of unassigned (z)OTUs among types.

Coniferous and broadleaved woodlands and extensively-managed grasslands displayed the highest proportions of functionally unassigned bacterial zOTUs (Fig. 8a, c), illustrating a bias in functional annotation between anthropic habitats and semi-natural environments. Extensive grasslands hosted the highest proportions (number and read counts) of unassigned fungal OTUs (Fig. 8b, d). We suggest that this result is due to the fact that other systems host more ectomycorrhizal fungi or pathogens that are relatively confidently assigned and we speculate that most of these unknowns are probably saprotrophs. It also highlights the need to improve functional annotation in

**Table 1 | Number of remaining sites at different normalisation thresholds and coverage of initial fungal richness**

| Threshold (read counts) | Number of sites | Coverage initial richness |
|---|---|---|
| 502 | 715 | 63.29% |
| 1000 | 668 | 80.23% |
| 1500 | 578 | 84.26% |
| 2000 | 488 | 83.43% |
| 2500 | 422 | 80.69% |
| 3000 | 348 | 75.12% |
| 3500 | 293 | 70.61% |
| 4000 | 237 | 63.55% |

**Table 2 | Comparison of fungal model performance for different normalisation thresholds**

| | Adjusted R² | | | | AIC | | | |
|---|---|---|---|---|---|---|---|---|
| | 502 | 1000 | 1500 | 2000 | 502 | 1000 | 1500 | 2000 |
| Observed richness | 0.1972 | 0.2225 | 0.2245 | 0.2177 | −145.19 | −155.33 | −135.07 | −103.13 |
| Shannon index | 0.1645 | 0.178 | 0.1874 | 0.1828 | −117.58 | −120.05 | −108.0863 | −85.73 |
| EcM | 0.6594 | 0.6254 | 0.6413 | 0.6425 | −977.65 | −857.39 | −707.23 | −584.35 |
| AMF | 0.2981 | 0.2872 | 0.3681 | 0.3726 | −239.26 | −215.27 | −251.49 | −216.66 |
| Fungal saprotrophs | 0.4473 | 0.4434 | 0.4719 | 0.4767 | −408.61 | −380.89 | −354.61 | −305.53 |
| Fungal plant pathogens | 0.3599 | 0.3453 | 0.3745 | 0.3817 | −303.11 | −274.03 | −256.71 | −227.03 |

Adjusted R² and AIC values for alpha-diversity and functional groups single-effect models based on sample-by-OTU tables normalised to a minimum of 502 read counts, 1000 read counts, 1500 read counts and 2000 read counts. EcM designate ectomycorrhizal fungi and AMF designate arbuscular mycorrhizal fungi.

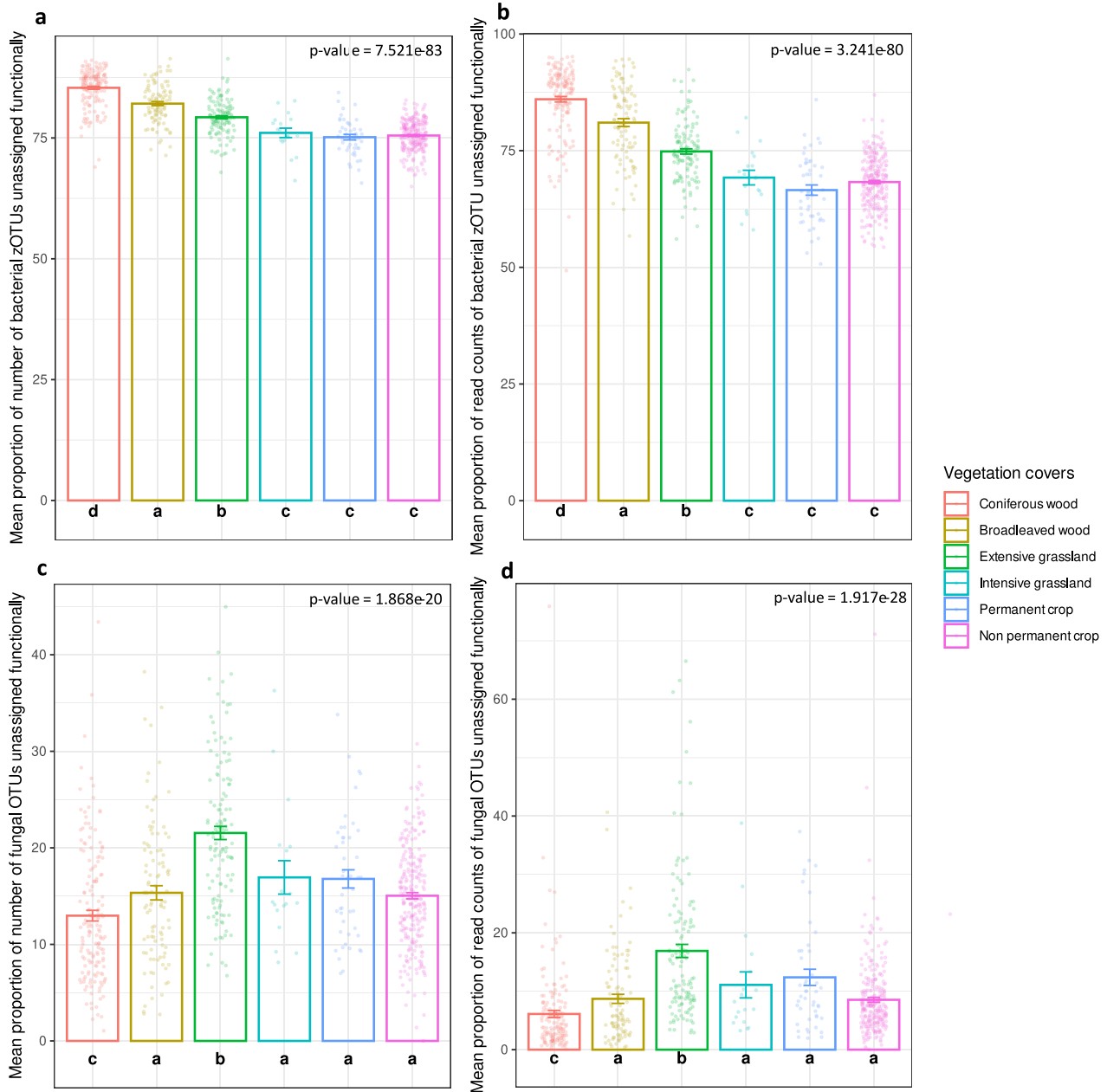

**Fig. 8 | Proportions (in percentages) of number and read counts of functionally unassigned bacterial zOTUs and fungal OTUs by vegetation cover type (mean ± SE). a** Proportions of number of functionally unassigned bacterial zOTUs. **b** Proportions of read counts of functionally unassigned bacterial zOTUs. **c** Proportions of number of functionally unassigned fungal OTUs. **d** Proportions of read counts of functionally unassigned fungal OTUs. Error bars represent standard error SE. Different letters correspond to a significant difference among proportions of (z)OTUs in compared vegetation cover types, and *p* value corresponds to the one obtained with a Kruskal-Wallis test testing the vegetation cover effect. Here, colours represent the different vegetation cover types and *n* = 715 total sites, with 160 belonging to coniferous woods, 99 to broadleaved woods, 128 to extensive grasslands, 18 to intensive grasslands, 46 to permanent crops and 264 to non-permanent crops sites. Source data are provided as a Source Data file.

grasslands to better assess functional group distribution among various vegetation cover types.

### Environmental variables
From the official LUCAS land cover nomenclature[94], three main land cover types were selected to conduct robust statistical analyses: cropland, grassland and woodland. A hierarchical system allowing the differentiation of these classes into finer cover types provided more insights on the potential role of vegetation cover. In particular, croplands were separated into permanent crops (e.g. fruit trees, olive groves and vineyards) and non-permanent crops (e.g. cereals and legumes).

Grasslands were separated into intensive grasslands (i.e. grassland identified at the time of the survey as a former cropland not cultivated for at least a year and not part of a crop-rotation, or an abandoned cropland) and extensive grasslands (i.e. permanent grassland covered by communities of grassland and grass-like plants and forbs). Woodlands were separated into coniferous or broadleaved forests.

In details, finer vegetation cover types were established based on EUROSTAT classification[94], as follows:
- Non-permanent crop (264 sites). The vegetation cover type "non-permanent crop" consisted of the following specific vegetation classes: Class B55 Temporary grassland (12 sites):

land occupied by temporary (and artificial) pastures, occupying the ground for at least one crop year and less than five years, the seeds being either pure or mixed grass, on cropland areas (i.e. making part of the crop rotation). Classes B54 Mixed cereals for fodder (2 sites), B53 Other legumes and mixture for fodder (8 sites), B52 Lucerne (5 sites), B45 Strawberries (1 site), B44 Floriculture and ornamental plants (1 site), B43 Other fresh vegetables (3 sites), B41 Dry pulses (6 sites), B37 Other non-permanent industrial crops (1 site), B35 Other fibre and oleaginous crops (1 site), B33 Soya (3 sites), B32 Rape and turning rape (15 sites), B31 Sunflower (16 sites), B23 Other root crops (2 sites), B22 Sugar beet (4 sites), B21 Potatoes (5 sites), B19 Other cereals (1 site), B18 Triticale (5 sites), B16 Maize (27 sites), B15 Oats (9 sites), B14 Rye (7 sites), B13 Barley (43 sites), B12 Durum wheat (12 sites), B11 Common wheat (75 sites).

- Permanent crop (46 sites). Classes B82 Vineyards (15 sites), B81 Olive groves (20 sites), B77 Other citrus fruits (1 site), B75 Other fruit trees and berries (5 sites), B74 Nuts trees (1 site), B73 Cherry fruit (1 site), B72 Pear fruit (1 site), B71 Apple fruit (2 sites).
- Intensive grasslands (18 sites). Class E30: Spontaneously re-vegetated surfaces: It consists of mostly agricultural land which has not been cultivated this year or the years before. It has not been prepared for sowing any crop this year. This class can also be found on clear-cut forest areas, industrial "brownfields", storage land and of course on abandoned or unused land etc. Main case is agricultural land not providing a crop during the entire year or abandoned earlier agricultural surfaces. It is occupied by spontaneous vegetation in case of set-aside arable land, with some tall herbs or weeds. This class applies as well for former grassland or hedge margins which are currently not used anymore but covered by tall herbs fringes. There might be some bare land pockets or crop residues and spontaneous re-grown crops of the before management period. Only surfaces which have not been deliberately sown and do not have any fodder crops like mixed cereals or are temporary grasslands classify for this land cover class.
- Extensive grasslands (128 sites). Class E10: Grassland with sparse tree/shrub cover (22 sites): Land predominantly covered by communities of grassland, grass-like plants and forbs including sparsely occurring trees (the tree canopy is between 5 and 10% and the total of the tree+shrub canopy is between 5 and 20% of the area). Class E20: Grassland without tree/shrub cover (106 sites): Land predominantly covered by communities of grassland, grass like plants and forbs without trees and shrub-land (density of tree+shrub canopy is less than 5%).
- Broadleaved forests (99 sites). Class C10: Broadleaved woodlands (82 sites): Areas with a tree canopy cover of at least 10% and composed of more than 75% of broadleaved species. Class C33: Other mixed woodland (17 sites): Mixed stands where less than 75% of the coniferous trees are spruce or pine trees.
- Coniferous forests (160 sites). Class C21: Spruce dominated coniferous woodland (32 sites): Coniferous stands where > 75% of the part of coniferous trees are spruce. Class C22: Pine dominated coniferous woodlands (51 sites): Coniferous stands where > 75% of the part of coniferous trees are pine species. Class C23: Other coniferous woodlands (14 sites): Coniferous stands where none of the previously mentioned coniferous species (pine or spruce) is represented > 75%. Class C31: Spruce dominated mixed woodlands (34 sites): Mixed stands where > 75% of the coniferous trees are spruce. Class C32: Pine dominated mixed woodlands (29 sites): Mixed stands where > 75% of the coniferous trees are pine.

Subsequently, the different vegetation cover types were ordered along a gradient of increasing land-use perturbation, from woodlands (less managed and less disturbed areas) to grasslands and croplands (highly-managed and more disturbed areas).

Vegetation cover served two distinct purposes in the analysis:

1. Assessment of land-use perturbation impact. Vegetation cover was considered as single factor including different types (e.g. coniferous forests, extensively-managed grassland and permanent crops). A multiple comparison of diversity metric values (or functional groups) among these types was carried out (e.g. bacterial and fungal α-diversity in coniferous forests compared to extensive grasslands and permanent crops).
2. Assessment of factors shaping microbial communities and functional groups. Vegetation cover was considered as a unique driver to be combined with and compared to other types of factors (i.e. climate and soil properties).

Quantified climatic variables were considered to broaden the spectrum of driving factors (i.e. temperature, rainfall, aridity) and capture the variability of the sampled locations (from the Mediterranean up to the Boreal regions). Climatic variables, averaged over the period 1970–2000, were obtained from the WorldClim database[98]. The climate values included were (i) monthly mean air temperature[98]; (ii) monthly aridity index (calculated based on the ratio between monthly total precipitation averaged over the period 1970–2000[98] and monthly potential evapotranspiration averaged over the same period[99]), where higher aridity index values indicated higher water availability and more humid conditions; (iii) iso-thermality: day-to-night temperatures oscillations relative to summer-to-winter (annual) oscillations[98]; (iv) temperature seasonality: amount of temperature variation over a given year (or averaged years)[98] (v) annual temperature range[98]; (vi) precipitation seasonality: variation in monthly precipitation totals over the course of the year (or averaged years)[98]. More details about bioclimatic variable are presented in O'Donnel & Inizio[100].

Soil chemical and physical properties were assessed for each site using International Organisation for Standardisation (ISO) methods. Those properties included: bulk density (0–20 cm, g.cm⁻³), clay and silt contents (%), coarse fragments (%), calcium carbonate content (g.kg⁻¹), extractable potassium content (mg.kg⁻¹), pH (in $H_2O$), available phosphorus content (mg. kg⁻¹) and the ratio between organic carbon (g.kg⁻¹) and total nitrogen content (g.kg⁻¹) (C:N ratio). Bulk density designated the weight of dry soil in a given soil volume, accounting for both the solid part and pore spaces of soil, and reflected soil compaction. Content of coarse fragments corresponded to the mineral particles not passing a 2-mm sieve and reflected soil texture[29]. ISO standards and references are available in Supplementary Table 6.

### Statistical analyses

All statistical analyses were performed using Rversion 4.2.1[101] and Rstudio version 2021.09.0[102]. All statistical analyses described hereafter were run separately for the bacterial and fungal datasets.

**Microbial diversity.** The observed OTU richness and Shannon index were calculated as a measure of bacterial and fungal α-diversity. Mean values taken by each α-diversity metric were compared across vegetation cover types using a Kruskal-Wallis and Pairwise Wilcoxon (with a Benjamini & Hochberg's correction) tests in order to explore land-use perturbation effect on α-diversity.

Multivariate ordinary least squares model[103] including all soil properties, climatic variables and vegetation cover together were used to assess the single effect of each of the three driver types (soil properties, climatic variables and vegetation cover) on α-diversity, for each metric separately. A selection of predictors was done using an Akaike Information Criterion (AIC) based stepwise feature selection (in both directions) on these models to reduce the number of variables included and to limit redundancy and multicollinearity in

variation partitioning analyses. Variation partitioning[104] was used on resulting models to assess the fraction of α-diversity variance explained uniquely by each type of drivers (i.e. soil properties, climatic variables and vegetation cover). Variable importance (VIP)[105], corresponding to the statistical significance of each considered variable with respect to its effect on the generated models, was used on these same models to establish a hierarchy between feature-selected numerical variables (soil properties and climate), and to compare their order between bacteria and fungi. Vegetation cover types were accounted for when investigating variable importance, but not visualised.

For each α-diversity metric separately, a stepwise feature selection was applied on a multivariate ordinary least squares model including all possible two-way interaction terms between drivers, i.e. all possible interaction terms between each soil property and vegetation cover type (soil properties × vegetation), each soil property and climatic variable (soil properties × climate), and each vegetation cover type and climatic variable (vegetation × climate). Variation partitioning was used to assess which type of interactions (soil properties × vegetation, soil properties × climate or vegetation × climate) led the models. Variable importance was used to establish a hierarchy between feature-selected interaction terms. Explained part (adjusted R-squared) and AIC values from the models were used as comparison metrics between the single-effect model and the interaction-effect model.

**Microbial phyla and classes.** At the community-level, (z)OTUs were grouped by phylum (or class) for each site. Total abundance of each taxonomic group was obtained by converting group counts to proportions. The ten most abundant phyla (or classes) corresponded to the ten phyla (or classes) that had the highest mean relative abundances in the dataset, across all sites. The mean relative abundances of each most abundant taxonomic group were obtained by vegetation cover type and converted to proportions to compare the groups among vegetation cover types on an equal basis. These steps allowed to set abundance of a phylum (or class) in each vegetation cover type to its mean proportion. Multiple comparison of mean relative abundances for each taxonomic group among vegetation cover types were established using a Kruskal-Wallis and Pairwise Wilcoxon posthoc tests.

The same pipeline was used at the functional group-level, investigating the phyla and classes (reduced to the ten most abundant classes when too numerous for visualisation). For that, entities (bacterial zOTUs or fungal OTUs) belonging to each functional group of interest were identified. For each functional group, previous analyses were performed on a *phyloseq*[106] object created with the sub-OTU table containing the entities belonging to the group, their taxonomic information and metadata information for the sites in which entities were detected.

**Community structure.** A dissimilarity matrix was calculated to assess the diversity in (z)OTUs between sites (β-diversity) using a Hellinger transformation of the sample-by-(z)OTU table and Bray-Curtis dissimilarity. A distance-based redundancy (dbRDA) analysis only including the vegetation cover was used to assess the effect of land-use perturbation on the β-diversity. Multilevel pairwise comparison with a Benjamini & Hochberg's correction was used to test the difference in community structure among communities belonging to different vegetation cover types.

Another dbRDA analysis tested the single effects of vegetation, soil properties and climatic variables together on the β-diversity after performing a stepwise feature selection on the standardised predictors. The significance level of each variable was evaluated with permuted ANOVA (999 permutations). Variation partitioning was used on the dissimilarity matrix in order to compare the fraction of

β-diversity variance explained uniquely by the feature-selected soil properties, climatic variables and vegetation cover. A multiple left-hand side (LHS) multivariate model with the feature-selected properties was run on the site scores of the first two axes of the dbRDA and variable importance was used on this model to establish a hierarchy between feature-selected numerical variables (soil properties and climate) on the first axis, and to compare their order between bacteria and fungi. Vegetation cover types were accounted for when investigating variable importance, but not visualised.

An ordination biplot was used to visualise interactions between drivers and the community β-diversity. Observed richness (or Shannon index), previously estimated, was added as isolines on the ordination plot to apprehend both α- and β-diversity relationships with the environmental variables simultaneously. A stepwise feature selection was applied on a dbRDA including all possible two-way interaction terms between type of drivers (i.e. between each soil property and vegetation cover type (soil properties × vegetation), each soil property and climatic variable (soil properties × climate), and each vegetation cover type and climatic variable (vegetation × climate). Variation partitioning was used to assess which types of interactions were the most important. Variable importance was used to establish a hierarchy between feature-selected interaction terms on the first axis, based on the multiple LHS multivariate model established on the site scores of the first two axes of the dbRDA accounting for selected interaction terms between drivers. Explained part (adjusted R-squared) and AIC values from the ordinations were used as comparison metrics between the dbRDA ordination based on single effects (after stepwise selection) and the dbRDA ordination based on interaction terms (after stepwise selection).

**Functional groups.** For each vegetation cover type, the mean proportion of (z)OTUs (weighted by their read counts) belonging to a given inferred bacterial or fungal functional group was determined. Subsequently, we assessed the impact of land-use perturbation level associated to each vegetation cover type on bacterial and fungal functional groups. Kruskal-Wallis and Pairwise Wilcoxon (with a Benjamini & Hochberg's correction) tests were performed to assess differences in mean proportions of functional groups among vegetation cover types.

The statistical analyses applied to α-diversity metrics (i.e. feature selection and multivariate ordinary least square models, variation partitioning, variable importance) were used to determine the most influencing set of environmental variables (soil properties, climate and vegetation cover) and their interactions driving the proportion of (z)OTUs belonging to each functional group. The relationship between microbial functional groups and the investigated drivers was visualised by means of ordination biplots.

**Complementary statistical analyses.** Each model assessing the impact of land-use perturbation and thus comparing a metric among vegetation cover types was performed on $n = 715$ total sites, with 160 belonging to coniferous woods, 99 to broadleaved woods, 128 to extensive grasslands, 18 to intensive grasslands, 46 to permanent crops and 264 to non-permanent crops sites. We set statistical significance at $p$ value $< 0.05$; if any other significance value was used, it was indicated in the corresponding table or figure.

In each multivariate ordinary least squares model testing the single- or interaction effects of vegetation cover, soil properties and climate, the explained variable was appropriately transformed in order to reach a normal distribution of model residuals (Supplementary Table 7). The sandwich estimator was used to calculate the variable importance when heteroscedasticity was observed in the model residuals. It was only used for the fungal functional groups, for both single- and interaction-effect models. Some important variables (or interaction terms) that were feature-selected by the multivariate ordinary

least squares models appeared as non-significant in one-way ANOVA tests. Those variables were kept in the models as they allowed to causally identify the parameters of other (significant) variables and removing them would bias the effect of the other variables. Except for β-diversity analyses, p values obtained with a permuted ANOVA (PER-MANOVA with 999 permutations and using Euclidean distance) were compared to the ones from a regular one-way ANOVA (see Supplementary Data file 2). Both p values were consistent and added as stars to the figures representing the variable importance of each variable in the models, but should be interpreted with caution for the models departing from ordinary least squares model assumptions. For beta-diversity analyses, p values from permuted ANOVA (999 permutations) performed on the ordinations were added as stars to the figures representing the variable importance of each variable in the models (see Supplementary Data file 2 for the exact p values).

Spatial autocorrelation was tested on all model residuals but none of the models showed strong geographical patterns that were not already accounted for by the set of environmental variables considered (Supplementary Fig. 17). Increasing semivariance with higher distances was attributed to the fact that the configuration of site locations is spatially constrained by the geographical shape of Europe. Elevation was either not selected by the feature selection step, or did not improve the model predictive power, representing quasi-null negative explained fraction in the variation partitioning analyses. As elevation is used to interpolate the WorldClim data on which are based the climatic variables used in the analyses, this variable was removed from the set of explaining variables in the models. The influence of seasons and sampling time was accounted for by including monthly climatic variables (i.e. monthly aridity index and monthly mean air temperature) which values for each site were established for the month during which the site was sampled.

**R packages**. All R packages used are listed with their version in the Reporting Summary associated to this paper. In details, *sf* package[107] was used to link every site to its corresponding biogeographical region when filtering the sample locations. Normalisation was performed using *SRS* package with Cmin argument equals to the minimum read counts for each filtered (bacterial or fungal) sample-by-(z)OTU table. Bioclimatic variables were obtained via the *dismo* package[108]. *microeco* package[109] permitted to relate bacterial zOTU to a set of potential ensured functions using the FAPROTAX database.

*phyloseq* package was used for the calculation of observed (z) OTU richness and Shannon diversity index, visualised using *ggplot2*[110] and *multcompView*[111] for the addition of the significance letters on the boxplots. Stepwise AIC for feature selection was performed via the *stepAIC* function from *MASS* package[112] in both directions. The significance level of each variable was evaluated with PERMANOVA using *adonis2* function from *vegan* package[113], 999 permutations and Euclidean distance. *phyloseq* package was also used for the calculation of the taxonomic groups at the phylum and classes within communities or functional groups. (z) OTUs were grouped by phylum (or class) for each site using *tax_glom* function. Total abundance of each taxonomic group was obtained by converting group counts to proportions with function *transform_sample_count*. The mean relative abundances of each most abundant taxonomic group were obtained by vegetation cover type using *merge_samples* function.

*vegan* package was used to calculate the dissimilarity matrix between any two pair of sites, to perform the dbRDA, feature selection with *ordistep* function and ordination plot inputs. Ordination plots were visualised with *ggordiplots*[114] and the *envfit* function from *vegan* package was used to fit the functional groups onto the plots. The significance level of each variable was evaluated with permuted

ANOVA using *anova.cca* function from *vegan* package and 999 permutations.

Kruskal-Wallis and Pairwise Wilcoxon (with a Benjamini & Hochberg's correction) tests were used in order to compare α-diversity or functional groups proportion among vegetation cover types (*kruskal.test* and *pairwise.wilcox.test* functions from *stats* package with p.adjust.method argument set to "BH"). Multilevel pairwise comparison with a Benjamini & Hochberg's correction was used to test the difference in community structure among communities belonging to different vegetation cover types (*pairwise.adonis* function from *pairwiseAdonis* package[115] with p.adjust.m argument set to"BH").

*vegan* package was also used for the variation partitioning outputs for both alpha and β-diversity analyses. *caret* package and *varImp* function[116] were used to investigate the variable importance between feature-selected predictors for both alpha and β-diversity analyses, *car* package and the *vif* function[117] were used to check for collinearity in the models.

*bestNormalize* package[118] was used to transform the explained variable and reach a normal distribution in the model residuals of the (single and interaction) models testing for the alpha-diversity and the functional groups (Supplementary Table 7). When heteroscedasticity was detected in the model residuals, a sandwich estimator was used to correct for it, via the *vcovHC* function from *sandwich* package[119,120] and *coeftest* function from *lmtest* package[121].

*geoR* package[122] was used to produce the variograms testing for spatial autocorrelation (Supplementary Fig. 17).

### Reporting summary
Further information on research design is available in the Nature Portfolio Reporting Summary linked to this article.

## Data availability
The raw data (DNA sequences) generated in this study have been deposited in the Sequence Read Archive (SRA) database under Bio-Project ID PRJNA952168. The sampling site metadata used in this study are available on the European Soil Data Centre (https://esdac.jrc.ec.europa.eu/content/soil-biodiversity-dna-bacteria-and-fungi). The data generated in this study are provided in the Supplementary Information, Supplementary Data 1 and 2. Source data are provided with this paper.

## Code availability
R scripts designed for data analyses are available on the European Soil Data Centre (https://esdac.jrc.ec.europa.eu/content/soil-biodiversity-dna-bacteria-and-fungi).

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

## Acknowledgements

We thank Leonidas Liakos (European Commission, Joint Research Centre (JRC), Ispra, Italy) for the map of LUCAS sites across the biogeographical regions (Fig. 1); Heidi Tamm (Institute of Ecology and Earth Sciences, University of Tartu, Estonia) for her detailed report on the protocols for DNA extraction, amplification and library preparation, used to develop the *DNA extraction and amplification* and *Metabarcoding library preparation* parts of the Methods section. The LUCAS Survey is coordinated by Unit E4 of the Statistical Office of the European Union (EUROSTAT). The LUCAS Soil sample collection is supported by the Directorate-General Environment (DG-ENV), Directorate-General Agriculture and Rural Development (DG-AGRI) and Directorate-General Climate Action (DG-CLIMA) of the European Commission. This work was realised in collaboration with the European Commission's Joint Research Centre under the Collaborative Doctoral Partnership Agreement No. 35594 with the University of Zurich. MvdH and FR acknowledge funding of the Swiss National Science Foundation (grant 310030_188799).

## Author contributions

M.L., MvdH., A.O. conceptualised the study. M.L., C.B., L.T., E.L., MvdH., A.O. designed the study. M.W.S., V.M., O.D., M.B. generated or processed the sequencing data. M.L. conducted statistical analyses (investigation, visualisation). F.R., P.P., A.J., L.T., MvdH. and A.O. supervised the work. M.L. and A.O. wrote the original draft. M.L., C.B., F.R., P.P., A.J., L.T., M.B., E.L., MvdH. and A.O. reviewed and edited the draft.

## Competing interests

The authors declare no competing interests.
