## [Peer Review File · Nature Communications]

Reviewers' Comments:

Reviewer #1:

Remarks to the Author:

The authors present an interesting large scale survey of terrestrial microbiomes, examining the relationship between community structure and putative function across Europe, and linking those to a comprehensive set of environmental parameters.

One concern is the incidence of fungal plant pathogens corresponding more with managed land. Might the database of known fungal pathogens be ag-biased? We might know more about the these plant pests in heavily researched systems, compared to 'wild' forests, though the pathogen load could still be just as high? I guess it might be a slippery slope to infer putative pathogens beyond the Faprotax assignments, but I am curious how you can account for this potential bias?

My other concern is spatial autocorrelation - I realize you did tests indicate no strong spatial pattern that's not already attributed to the env parameters, but those would likely also have the ~same geographic pattern? Just seems like the coniferous-associated microbiomes clump in the ordination as they do spatially within Scandanavia? I guess the other vegetation types are more mixed across the continent.

Finally, from a biogeochemistry standpoint, I wonder if what is presented here is actually function, or instead putative/inferred functional potential? Because the function here is inferred by the community data, I would contend that maybe this is more functional potential or putative function?

Minor point - can you stick with one color scheme - I kept thinking the green points at the zenith of the dbRDA were coniferous.

Line-by-line

114 - Oof, what happened to the precipitous drop in OTUs from 'total' listed here, compared to the rarefied numbers in L513, especially for Fungi? Could you simply drop the one sample pulling these numbers down? Or, instead use data transformation (see Greg Gloor's paper for recommendations on why that might be better than rarefactions - <https://www.frontiersin.org/articles/10.3389/fmicb.2017.02224/full>)

561 - No soil moisture? Several studies (incl. [pnas.org/doi/full/10.1073/pnas.1620811114](https://doi.org/10.1073/pnas.1620811114)) would highlight the importance of historical precipitation, but contemporary soil moisture may still be good to include. If not directly calculating GWC from the collected soil samples, at least inferred from near recent rainfall data? I realize this is a continental scale dataset, with a many long-term parameters input into the model, but what if something more dynamic is the primary determinant of community structure and putative function?

Reviewer #2:

Remarks to the Author:

The manuscript "Vegetation, climate and soil properties drive soil bacterial and fungal communities and associated functions across Europe" by Labouyrie et al., is taking an advantage of substantial soil sample collection generated through Land Use/Cover Area frame Survey (LUCAS) initiative. Currently LUCAS database contains 351293 observations at 651780 unique locations for 106 variables, which do not include soil microbiome profiles. The complementation of LUCAS dataset with the information about soil microbiome diversity and structure will be extremely valuable for scientific community. The authors did a nice job looking for correlation between environmental factors and bacterial and fungal community diversity and functions.

However, I have a major concern about the dataset used in the study. Looking at rarefaction curves (S2, Fig.1): The fig 1 does not provide a good information about samples with low counts. But just looking at bacterial dataset (fig. 1) it is apparent the at the cutoff of 35,849 16S reads most of the samples are far from species saturation, so they might not be sufficiently sequenced to

represent its identity. For the figure 1, it would be useful to decrease the maximum depth during rarefaction process to 1,000,000 to see the behavior of samples at the low depth. It would be also useful to calculate Shannon and OUT Richness parameters during rarefaction for each sample and present it as supplementary information. Based on these data the authors could prove that the normalization to 35,849 16S reads did not compromised the data. I have the same concerns about fungal dataset.

I had some questions about bacterial functional annotation, how they were grouped into broad functional groups, but I could not find this data in supplementary information. Table with bacterial and fungal OTUs and their taxonomic and functional annotation should be included in the submission.

Data availability: Please make sure that the data are available on LUCAS website. I could not find it.

Reviewer #3:

Remarks to the Author:

Labouyrie and co-workers described a survey of the soil microbial communities identified in over 700 samples derived from eight European biogeographical regions. By using high-throughput sequencing of phylogenetic markers and computational predictions, they determined the taxonomic composition and functions of the microbiota of each sample. Next they used a set of 15 co-variables, representing edaphic factors and climatic variables at the sites of sampling, to identify putative drivers of microbial diversification. The latter were weighed for their capacity of predicting communities' properties as individual factors or in combinations. Finally the authors proposed how to capitalise on the generated results to inform actions aimed at nurturing soil biodiversity at continental scale.

The distinctive feature of the manuscript is the sampling and analysis of soil microbiota data at continental scale. It is not difficult to imagine that this dataset may become a reference one for future investigations. Yet, there are several aspects of this investigation requiring further clarifications.

#1. The depicted functional annotation is in fact a prediction and, as such, as good as the database used. For instance, the assumptions of the method used may not adapt to phylogenetically related taxa, with identical or nearly identical 16S rRNA gene, capable of performing distinct functions (e.g., PMID: 35210578). For this reason, the conclusion inferred using the 'functional annotation' have to be supported by experimental data. My suggestion would be a metagenomic survey using the available DNA of a few selected samples to validate (at least some of) the functional predictions (e.g., PMID: 23236140).

#2. It is highly unusual for a microbiota manuscript not having a taxonomic breakdown of the sequenced samples, at least at high ranks (e.g., Phylum or Class) according, for example, to vegetation cover. This would allow readers to compare the obtained results with other soil microbiota survey (e.g., Extended Figure 1 in PMID: 30069051).

#3. The description of sequencing data processing requires further details. For instance, the fate of technical replicates is unclear: did they confirm a lack of significant library-to-library variation? If yes I'd invite the authors to include an additional supplementary figure to illustrate this critical point. If no, how did the authors correct for that? Likewise, the authors should include the minimum PHRED score used to retain high-quality sequences and the threshold for taxonomic assignment (e.g., 80% confidence?). Finally it is unclear what was the fate of OTUs not classified at lower ranks (e.g., an OTU classified at a family but not a genus level): were those retained for the functional prediction or excluded? If the latter, have the authors corrected for potential biases, e.g., a scenario in which cropland had more classified OTUs at genus level versus woodland.

#4. While conceptually correct, the use of OTUs and the RDP database may limit the re-usage of the dataset. ASVs and SILVA database would have been a more up-to-date approach.

#5. I was somehow lost in the discussion: if the overarching aim is gaining accuracy in predicting

'healthy' versus 'degraded' soils, I think is important factoring in plant responses. For instance, it is unclear whether the croplands yielded equally well or not: in absence of this data it would be impossible discriminating predictors of microbial characteristics associated with a fertile soil. I believe that this limitation should be at least acknowledged in the discussion.

#6. In addition, there are concepts that are difficult to reconcile with the biology of plants and microbes in soil: (L. 353-55) how can obligate biotrophs be predicted by edaphic factors only?

#7. Likewise, some of the proposed actions appear impractical: (L.411-12) varying soil pH will likely alter mineral availability for plant uptake, hence the other soil variables may not be kept "fixed".

#8. The lack of access to raw sequences, sampling information as well as scripts used to produce figures and statistical analyses prevents these critical data to be properly evaluated. I suggest to release this information promptly and, whenever possible, by using open-access established repositories (e.g., ENA-EBI, Zenodo, GitHub...).

#END

Dear all,

We thank you for revising our manuscript and for the many helpful suggestions and comments that helped us to improve the overall content of the manuscript. Please find our point-by-point replies below in orange.

Sincere regards on behalf of all the authors,

Alberto Orgiazzi

Marcel van der Heijden

--

Reviewer #1 (Remarks to the Author):

The authors present an interesting large scale survey of terrestrial microbiomes, examining the relationship between community structure and putative function across Europe, and linking those to a comprehensive set of environmental parameters.

Thank you for the overall positive overview.

#1. One concern is the incidence of fungal plant pathogens corresponding more with managed land. Might the database of known fungal pathogens be ag-biased? We might know more about these plant pests in heavily researched systems, compared to 'wild' forests, though the pathogen load could still be just as high? I guess it might be a slippery slope to infer putative pathogens beyond the Faprotax assignments, but I am curious how you can account for this potential bias?

Thank you very much for pointing out this limitation. We agree with this and have now included a statement in the Discussion to cover this point (see lines 470 to 473). FungalTraits and FAPROTAX are the databases used for functional annotations of fungal and bacterial data, respectively. As you mentioned, both databases do present potential biases in terms of functional annotation among ecosystem types, but we could not account properly for this potential bias and have not found any published studies that cover this. However, our observations on fungal pathogens predominating in agricultural lands join previous findings by Le Provost et al. (2021) (PMID: 34168127) that also investigated similar functions.

#2. My other concern is spatial autocorrelation - I realize you did tests indicate no strong spatial pattern that's not already attributed to the env parameters, but those would likely also have the ~same geographic pattern? Just seems like the coniferous-associated microbiomes clump in the ordination as they do spatially within Scandinavia? I guess the other vegetation types are more mixed across the continent.

Thank you for raising this concern, that we hope to address with following explanations. The spatial autocorrelation of the independent variables does not bias the outcome of regression model. As a matter of fact, the correlation (including the spatial one) between dependent and independent variables is what allows modelling them statistically. So, forest type will follow a more or less well-defined pattern due to climate and geology, but that does not mean that a distribution model based on those two factors is flawed.

As spatial autocorrelation refers to the correlation of the dependent variable (e.g., α -diversity metrics, community composition) with itself, not to the autocorrelation within one of the independent variables, we needed to test if there was a residual spatial autocorrelation of the dependent variable not accounted for in the regression models.

A way to verify if spatial autocorrelation exists due to the distribution of the vegetation cover types across Europe is, e.g., to i) extract the site scores on the two axes of the dbRDA ordination only testing for vegetation cover effect (i.e., Fig. 2 of the manuscript), ii) consider those site scores as variables to explain by the vegetation cover, and iii) perform a variogram to investigate spatial autocorrelation. The variograms below (Figure 1) indicate a very small residual spatial autocorrelation. However, the variograms associated to our full models (including vegetation cover, soil properties and climate) in Supplementary Methods 8 (Fig. 2) do not display patterns of spatial autocorrelation, indicating that the variables included in the models captured also the residual spatial autocorrelation of our

dependent variables. Hence, our full models already account for the spatial distribution of forests by considering the correlation between vegetation cover type and the microbiomes in the regression.

Figure 1: Variograms of models based on the site scores of the first two axes of the ordination (dbRDA) testing for the effect of vegetation cover only on bacterial or fungal community composition.

#3. Finally, from a biogeochemistry standpoint, I wonder if what is presented here is actually function, or instead putative/inferred functional potential? Because the function here is inferred by the community data, I would contend that maybe this is more functional potential or putative function?

We agree with this. The functions presented here are indeed potential (inferred) functions established based on (z)OTUs taxonomy using FungalTraits and FAPROTAX functional databases. This point has been now clearly mentioned in the Introduction section (line 94) and reminded along the manuscript (in the Results and Discussion sections). In addition, limitations to this approach are also discussed in the manuscript (lines 466 to 489) as requested by the other reviewers.

#4. Minor point - can you stick with one color scheme - I kept thinking the green points at the zenith of the dbRDA were coniferous.

Thank you, done: in figure 1, the map has been updated with the same colors than used in the plots. The green part of the land-use intensification gradient has been softened to avoid confusion and not to associate a green color to coniferous forests.

Line-by-line

#5. 114 - Oof, what happened to the precipitous drop in OTUs from 'total' listed here, compared to the rarefied numbers in L513, especially for Fungi? Could you simply drop the one sample pulling these numbers down? Or, instead use data transformation (see Greg Gloor's paper for recommendations on why that might be better than rarefactions - <https://www.frontiersin.org/articles/10.3389/fmicb.2017.02224/full>)

Thank you for raising this concern and proposing alternatives. For fungi, due to the composite nature of the dataset, we only considered samples with at least 500 total read counts, discarding 37 sampling sites and obtaining 25,962 OTUs across 715 sites. The total read counts in the smallest sample was 7 in the initial dataset. As explained in the answer to comment #1 from reviewer #2, selecting a 502-threshold is a compromise between covering enough initial richness without losing too many samples in the process, in order to investigate a representative ensemble of sites in terms of vegetation cover, soil and climatic conditions across Europe. This approach does not compromise the conclusions and quality of results, as we have performed additional analyses with different thresholds and found similar outputs (see Supplementary Method 3). It also permits to include a large number of sites in our analyses in which comparing bacterial communities.

Also, the method used in our analyses is the Scaling with Ranked Subsampling (SRS) as proposed by Beule and Karlovsky (2020 - PMID: 32832266). This approach does not consist in a random subsampling without replacement as the commonly used rarefaction method from McMurdie and Holmes (2014 - PMID: 24699258). A comprehensive demonstration of SRS's advantages can be found in Figures 1 and 2 of the paper from Beule and Karlovsky, making it a suitable method for the normalization of microbiome data. Therefore, we corrected the term « rarefaction » by « normalization » in our manuscript to avoid this confusion.

Over preliminary analyses we also tested the CLR and Aitchison transformations (as proposed by Greg Gloor et al.) but such alternatives are beyond state-of-the-art for large datasets and are prone to overfitting and distortion of the results (see, for instance, Martino et al., 2019 - PMID: 30801021). That is why we finally selected the SRS method to proceed with downstream analyses.

#6. 561 - No soil moisture? Several studies (incl. [pnas.org/doi/full/10.1073/pnas.1620811114](https://doi.org/10.1073/pnas.1620811114)) would highlight the importance of historical precipitation, but contemporary soil moisture may still be good to include. If not directly calculating GWC from the collected soil samples, at least inferred from near recent rainfall data? I realize this is a continental scale dataset, with a many long-term parameters input into the model, but what if something more dynamic is the primary determinant of community structure and putative function?

Thank you for this suggestion. Following your point, we used the soil moisture data from the NASA-USDA Enhanced SMAP Global soil moisture, providing information across the globe at 10-km spatial resolution. We obtained soil moisture (ssm) and subsurface soil moisture (susm) value for each sampling point, for the day (3-day-composite information), week and month at which each site was sampled in 2018. We chose to investigate the daily ssm and daily susm as representative components of the soil moisture, as daily data better accounted for its temporal dynamic within a month or week (see Figure 2 below). We reran the feature-selection models including all soil properties, vegetation, climate, and soil moisture (daily ssm + daily susm), and investigated if the soil moisture was selected as important, checked the R^2 , AIC, and variation partitioning outputs of the models. Soil moisture was kept as an explaining variable in the models in the case of fungal observed OTU richness (daily susm only), bacterial chemoheterotrophs (daily ssm and daily susm), bacterial N-fixers (daily ssm and daily susm) and fungal saprotrophs (daily ssm and daily susm). By comparing the overall models where soil moisture was included in the feature selection step to models that did not include it, we found that soil moisture improved the variance explained by 0.02% maximum for equivalently parsimonious models (based on AIC values - Table 1 below) and held between 0.002% and 0.006% of unique variance in the models (Figure 3 below). In case, we also tested the impact of weekly and/or monthly soil moisture, and obtained similar observations. Also, we replaced missing data present in our dataset with substituted values (imputation) (see lines 165 to 199 of the R script entitled « 5_Analyses_models_alpha_diversity_functions_16S_ITS.R ») including all soil moisture data (ssm and susm, daily, monthly and weekly) as additional information for a more robust imputation. From this, we propose to keep the soil

moisture information as part of the initial LUCAS metadata for future users to access it, but not to include soil moisture in the present manuscript as this information does not drastically improve the performance of our models.

We hypothesize that short-term soil moisture has a very limited impact on bacterial and fungal communities (and associated functional groups) because:

- DNA data were used in the analyses, thus we also study DNA of potentially dried-up quiescent organisms, dead organisms, that won't react to short-term changes in soil moisture.
- Soil moisture changes are not so big as to influence the overall diversity: the long-term climate is what impacts the diversity, with changes in the microbial communities that are more pronounced after longer periods of climatic changes (see, for instance, Seaton et al 2021).

Figure 2: Daily soil moisture (ssm) (a) and subsurface soil moisture (susm) (b) for one LUCAS point over 2018, and visualization over the month of June 2018 to illustrate the temporal dynamic of ssm (c) and susm (d).

Table 1: Adjusted R² and AIC values for models excluding or including soil moisture as a variable in the feature selection step.

		Models without soil moisture		Models with soil moisture	
		Adjusted R ²	AIC	Adjusted R ²	AIC
Observed richness	Bacteria	0.4889	-463.41	0.4887	-463.09
	Fungi	0.2035	-149.79	0.2058	-149.93
Shannon index	Bacteria	0.5437	-543.47	0.5435	-543.23
	Fungi	0.1679	-119.56	0.1679	-119.56
B-diversity	Bacteria	0.3734	3573.29	0.3937	3572.09
	Fungi	0.1279	4051.07	0.1293	4051.87
Chemoheterotrophs	Bacteria	0.6419	-719.79	0.6443	-721.64
N-fixers	Bacteria	0.3258	-268.05	0.3299	-270.45
Human pathogens	Bacteria	0.3536	-298.15	0.3536	-298.15
Ectomycorrhiza	Fungi	0.6591	-976.97	0.6591	-976.97
Arbuscular mycorrhiza	Fungi	0.2957	-236.83	0.2957	-236.83
Saprotrophs	Fungi	0.4447	-405.22	0.4463	-406.27
Plant pathogens	Fungi	0.3581	-301.19	0.3581	-301.19

Figure 3: Variation partitioning of models keeping soil moisture as an important variable after feature selection step.

Reviewer #2 (Remarks to the Author):

The manuscript "Vegetation, climate and soil properties drive soil bacterial and fungal communities and associated functions across Europe" by Labouyrie et al., is taking an advantage of substantial soil sample collection generated through Land Use/Cover Area frame Survey (LUCAS) initiative. Currently LUCAS database contains 351293 observations at 651780 unique locations for 106 variables, which do not include soil microbiome profiles. The complementation of LUCAS dataset with the information about soil microbiome diversity and structure will be extremely valuable for scientific community. The authors did a nice job looking for correlation between environmental factors and bacterial and fungal community diversity and functions.

Thank you for the overall positive overview.

#1. However, I have a major concern about the dataset used in the study. Looking at rarefaction curves (S2, Fig.1): The fig 1 does not provide a good information about samples with low counts. But just looking at bacterial dataset (fig. 1) it is apparent the at the cutoff of 35,849 16S reads most of the samples are far from species saturation, so they might not be sufficiently sequenced to represent its identity. For the figure 1, it would be useful to decrease the maximum depth during rarefaction process to 1,000,000 to see the behavior of samples at the low depth. It would be also useful to calculate Shannon and OUT Richness parameters during rarefaction for each sample and present it as supplementary information. Based on these data the authors could prove that the normalization to 35,849 16S reads did not compromised the data. I have the same concerns about fungal dataset.

Thank you very much for raising these concerns. We have now addressed this point and performed analyses for different thresholds for the fungal data (see Supplement Method 3). For the bacterial data, following a comment from reviewer #3, we replaced the 99%-OTUs 16S data by zOTUs (or ASVs, 100% similarity threshold), but used the same normalization approach. For the normalization step, we used the SRS method (Beule and Karlovsky, 2020 - PMID: 32832266), that is not based on a random sampling without replacement as proposed in the rarefaction method by McMurdie and Holmes (2014 - PMID: 24699258). We replaced the term « rarefaction » by « normalization » in our manuscript to avoid this confusion. Authors of the SRS method advise to use the total read counts of the smallest sample as threshold to which one should normalize the sample-by-(z)OTU table. That is what we did in the analyses, selecting the minimum total number of read counts for bacterial data, now 40,109 read counts (before it was 35,849). By choosing this sequencing depth, we kept 99.99% (before: 99.78%) of the initial richness across 715 sites, covering the behavior of samples at the low depth as well. Below is a zoom-in on SRS normalization curves at the 40,109-threshold (cropped at 100,000 reads for better visualization). We replaced the bacterial curves from Supplementary Method 3 by the visualization below (see new Supplementary Method 3).

Different thresholds can be tested using the SRS.shiny.app (developed by the same authors), that permits to assess the number of sites and OTUs that would be considered at different sequencing depths. In addition, thanks to this comment, we spotted an anomaly in our input data for the bacterial 99%-OTUs and were able to correct for it before running all analyses again. We obtained similar conclusions than previously presented.

As of fungal data, we first discarded all samples with less than 500 total read counts due to the highly compositional nature of the data. After this filtering step, we again chose the sample with the lowest total read counts as

sequencing depth to which normalize the data (502 read counts), keeping 63.29% of the initial richness and discarding 37 samples in the process. By doing so, we covered the smallest samples from our dataset (see SRS curves below, also added to new Supplementary Method 3). Selecting higher thresholds discarded a non-negligible number of sites (Table 2 below), that removed valuable information (e.g., combination of vegetation cover and bioclimatic context) from the dataset and led to unbalanced representations of the vegetation cover, soil and climatic conditions found across Europe. We now provide the observed richness and Shannon index values found at different thresholds as Supplementary Data file 1. Correlation matrices evidence that for the same not-discarded sites (at different thresholds, hence, up to 237 sites for a 4,000-reads threshold - Table 2), observed richness values were correlated to each other with a r^2 ranging 0.92-1.00 (Figure 4 below) and Shannon index values correlated with $r^2 > 0.9999$.

We also ran our analyses with a 1,000-threshold (including 668 sites), a 1,500-threshold (including 578 sites) and a 2,000-threshold (including 488 sites), that represent optima for richness coverage (Table 2), and compared the results both quantitatively and qualitatively among thresholds (Table 3). Table 3 and Figure 5 below show a fraction of our results: the observed richness and Shannon index are equivalently well predicted at the different normalization thresholds (adjusted R^2 and AIC) and we retrieved the same main conclusions as the ones presented in our manuscript (e.g., fungal α -diversity mainly shaped by vegetation cover). Similar observations were done for the β -diversity (adjusted R^2 of 0.1279 at 502-threshold, 0.1088 at 1,000-threshold, 0.1339 at 1,500-threshold and 0.1353 at 2,000-threshold).

Thus, the main conclusions of the manuscript do not change when taking different thresholds (this point is now addressed in Supplementary Method 3). As our aim is to determine patterns and propose conclusions at large scale, we chose to keep the 502-threshold for normalization of the data. Analyzing data from a higher number of sites also permits to compare both bacterial and fungal communities at a large scale, from the exact same sampling sites (and thus, exact same environmental conditions), without losing more information for the bacterial dataset that was already cut by 37 sites. We added these justifications and below tables 2 and 3 to Supplementary Method 3.

Table 2: Number of remaining sites at different normalization thresholds and coverage of initial fungal global richness.

Threshold (read counts)	Number of sites	Coverage of initial richness
502	715	63.29%
1,000	668	80.23%
1,500	578	84.26%
2,000	488	83.43%
2,500	422	80.69%
3,000	348	75.12%
3,500	293	70.61%
4,000	237	63.55%

Figure 4: Correlation matrix among fungal observed OTU richness values calculated for 237 common LUCAS sites at different normalization thresholds. The numbers next to “Observed_richness” indicate the number of read counts to which normalization was performed.

Table 3: Adjusted R^2 and AIC values for alpha-diversity and functional groups single-effect models based on sample-by-OTU tables normalized to a minimum of 502 read counts, 1,000 read counts, 1,500 read counts and 2,000 read counts.

	Adjusted R^2				AIC			
	502	1,000	1,500	2,000	502	1,000	1,500	2,000
Observed richness	0.1972	0.2225	0.2245	0.2177	-145.19	-155.33	-135.07	-103.13
Shannon index	0.1645	0.178	0.1874	0.1828	-117.58	-120.05	-108.09	-85.73
EMF	0.6594	0.6254	0.6413	0.6425	-977.65	-857.39	-707.23	-584.35
AMF	0.2981	0.2872	0.3681	0.3726	-239.26	-215.27	-251.49	-216.66
Fungal saprotrophs	0.4473	0.4434	0.4719	0.4767	-408.61	-380.89	-354.61	-305.53
Fungal plant pathogens	0.3599	0.3453	0.3745	0.3817	-303.11	-274.03	-256.71	-227.03

Figure 5: Comparison variation partitioning for observed richness (left) and Shannon index (right) values calculated from a sample-by-OTU table normalized at a threshold of (a) 502 read counts, (b) 1,000 read counts, (c) 1,500 read counts and (d) 2,000 read counts.

#2. I had some questions about bacterial functional annotation, how they were grouped into broad functional groups, but I could not find this data in supplementary information.

Thank you for pointing out this gap. The grouping information is now added as a new Supplementary material S4, and the table for bacterial annotation with FAPROTAX database is added as an output to the R scripts and raw data

(.csv document "Bacterial_functional_annotation_FAPROTAX_for_16S_zOTUs.csv") on *Nature Communications'* Google Drive.

#3. Table with bacterial and fungal OTUs and their taxonomic and functional annotation should be included in the submission. Data availability: Please make sure that the data are available on LUCAS website. I could not find it.

All raw data, including OTU tables, and scripts are provided on *Nature Communications'* Google Drive. Additionally, these data will be made available in the European Soil Data Centre (ESDAC - <https://esdac.jrc.ec.europa.eu/>) as soon as the article is published. Please note that, following European Commission's open access policy, ESDAC already hosts all LUCAS-related datasets: <https://esdac.jrc.ec.europa.eu/resource-type/datasets>.

--

Reviewer #3 (Remarks to the Author):

Labouyrie and co-workers described a survey of the soil microbial communities identified in over 700 samples derived from eight European biogeographical regions. By using high-throughput sequencing of phylogenetic markers and computational predictions, they determined the taxonomic composition and functions of the microbiota of each sample. Next they used a set of 15 co-variables, representing edaphic factors and climatic variables at the sites of sampling, to identify putative drivers of microbial diversification. The latter were weighed for their capacity of predicting communities' properties as individual factors or in combinations. Finally the authors proposed how to capitalise on the generated results to inform actions aimed at nurturing soil biodiversity at continental scale.

The distinctive feature of the manuscript is the sampling and analysis of soil microbiota data at continental scale. It is not difficult to imagine that this dataset may become a reference one for future investigations. Yet, there are several aspects of this investigation requiring further clarifications.

Thank you for the overall positive overview.

#1. The depicted functional annotation is in fact a prediction and, as such, as good as the database used. For instance, the assumptions of the method used may not adapt to phylogenetically related taxa, with identical or nearly identical 16S rRNA gene, capable of performing distinct functions (e.g., PMID: 35210578). For this reason, the conclusion inferred using the 'functional annotation' have to be supported by experimental data. My suggestion would be a metagenomic survey using the available DNA of a few selected samples to validate (at least some of) the functional predictions (e.g., PMID: 23236140).

Thank you for pointing out these limitations. The approach we used follows the work done, for instance, by Sansupa et al. 2021 who found that more than 97% of the FAPROTAX assigned OTUs for soil bacteria have previously been detected and potentially performed functions in agricultural and forest soils. The use of the FAPROTAX database also joins some recent works investigating potential bacterial functions in croplands (e.g., PMID: 35688828) or in bulk and rhizosphere soils (e.g., PMID: 35149704). Bearing this in mind, and in order to be more precise, as also requested by the Editor, we now refer to "inferred/potential functions" throughout the manuscript. Furthermore, we discussed the limitations of our approach in the manuscript (lines 466 to 489).

Finally, more than metagenomics, which is still based on DNA, thus showing the potential functions carried out by microbial communities, we think that metatranscriptomics and metaproteomics data would better allow to validate functional assignments. In order to partially fill in this gap, we are now considering the possibility to include RNA analysis in future LUCAS Soil Biodiversity surveys.

#2. It is highly unusual for a microbiota manuscript not having a taxonomic breakdown of the sequenced samples, at least at high ranks (e.g., Phylum or Class) according, for example, to vegetation cover. This would allow readers to compare the obtained results with other soil microbiota survey (e.g., Extended Figure 1 in PMID: 30069051).

Thank you for this valuable point. We have now added the mean relative abundance of most abundant phyla and classes for bacteria and fungi at community level (see Supplementary Table 2). Mean relative abundance of all phyla and classes (or most abundant classes when more than ten classes were found) for bacterial and functional groups were added as Supplementary Tables 3 (bacteria) and 4 (fungi). This information is also described in the manuscript (lines 132 to 165 of the Results, lines 289 to 291 of the Discussion and lines 683 to 700 of the Methods). Taxonomic information breakdown to vegetation cover was added to the manuscript (lines 158 to 165 of the Results) and corresponding plots were added as Supplementary Fig. 1 to 5.

#3. The description of sequencing data processing requires further details. For instance, the fate of technical replicates is unclear: did they confirm a lack of significant library-to-library variation? If yes I'd invite the authors to include an additional supplementary figure to illustrate this critical point. If no, how did the authors correct for that? Likewise, the authors should include the minimum PHRED score used to retain high-quality sequences and the threshold for taxonomic assignment (e.g., 80% confidence?). Finally it is unclear what was the fate of OTUs not classified at lower ranks (e.g., an OTU classified at a family but not a genus level): were those retained for the functional prediction or excluded? If the latter, have the authors corrected for potential biases, e.g., a scenario in which cropland had more classified OTUs at genus level versus woodland.

Thank you for mentioning those different points. We have now provided further information about the methodology used in the Supplementary Methods 2 and 4. Given the size of the sampling design (samples from 715 sites, both for bacteria and fungi), it was not feasible to collect and process multiple samples from individual sites to assess within site or library-to-library variation. Thus, we cannot separate this from the overall variation. However, samples were processed randomly and not in batches, which leads to the library-to-library variation just being added to the random error variance. Hence, it does not result in systematic biases (i.e., wrong interpretations) but rather just reduces statistical power of our models.

We added further information about bacterial and fungal taxonomic assignment to Supplementary Method 2. We added taxonomic levels to which functional annotation was performed in Supplementary Method 4. We investigated the proportion of number of (z)OTUs functionally unassigned among vegetation cover types and the proportion of read counts associated to them, and added this work in Supplementary Method 4. We mentioned the number of (z)OTUs functionally unassigned in the Results section of the manuscript (lines 144 to 146). The proportions of functionally unassigned (z)OTUs are unbalanced between ecosystem types, however, due to methodological constraints we could not correct for this annotation bias in our analyses. Also, we did not find any published studies that either mentioned it or correct for it. Therefore, we added this point to the limitations in the Discussion part of the manuscript (lines 470 to 472).

Additionally, thanks to this comment, we spotted an error in our scripts that we corrected in this revised version. Indeed, the proportions calculated for bacterial groups (chemoheterotrophs, N-fixers and pathogens) were previously based on the number of OTUs and not their read counts. Hence, former results presented in Figure 2 of our manuscript indicated, for example, a higher number of potentially pathogenic OTUs in more anthropic habitats, but when weighted them by their read counts, results now show that potential bacterial pathogens' relative abundance (in terms of read counts) is higher in coniferous woods. Such correction also influenced part of associated results that we updated accordingly (e.g., variation partitioning - line 198; soil properties and climatic conditions in which bacterial N-fixers and pathogens can be found - lines 207 to 211 and 216 to 222 of the revised manuscript). This error was due to an inconsistency in nomenclature among *microeco* R package versions that we did not spot and update accordingly when using the most recent version. To us, this correction was necessary as weighting (z)OTUs by their read counts better represent the functional groups loads, and better account for cases in which numerous (z)OTUs only represent few read counts. Fungal groups proportions remain unchanged and were correctly weighted in the previous version of the manuscript.

#4. While conceptually correct, the use of OTUs and the RDP database may limit the re-usage of the dataset. ASVs and SILVA database would have been a more up-to-date approach.

Thank you for raising those points. We updated our analyses with zOTUs (zero-radius OTUs, i.e., ASVs) for the bacterial 16S data as you requested. As a result, we detected less zOTU sequences compared to the 99%-OTU sequences used earlier, as a greater number of sequences had very low counts and were filtered out, and a greater

number of remaining sequences were not taxonomically assigned. We did not update our analyses with ASVs for fungi. ASVs are not considered suitable for full-length ITS sequences and are not optimal due to random PCR errors and the presence of multiple/highly similar copies of the ITS region in eukaryote genomes (Lindner et al. 2013 - PMID: 23789083). The use of ASVs increases the elimination of taxa that are both rare and phylogenetically unique (Joos et al. 2020 – PMID: 33092529). Finally, the analyses from which the initial fungal dataset was produced, showed that the ASV approach discarded more sequences compared to the OTU approach.

On the choice of RDP over SILVA database, it has been argued that one out of five predictions in large databases (like SILVA) are wrong, while smaller databases, as the RDP, are more reliable (a good overview is given here with all related references: https://drive5.com/usearch/manual/faq_tax_db.html).

#5. I was somehow lost in the discussion: if the overarching aim is gaining accuracy in predicting 'healthy' versus 'degraded' soils, I think is important factoring in plant responses. For instance, it is unclear whether the croplands yielded equally well or not: in absence of this data it would be impossible discriminating predictors of microbial characteristics associated with a fertile soil. I believe that this limitation should be at least acknowledged in the discussion.

We thank you for raising this comment that permit to better present the possible limits of the present study. We added this limitation to the Discussion (lines 460 to 464). Despite the large number of metadata collected by LUCAS Soil survey, quantified information on plants (e.g., plant community richness and structure, yield for crops) are still poor. We are fully aware that this lack might have partially affected our dataset as it could explain part of the unexplained variance of our models. Also, we added a concluding paragraph to summarize the main findings of our study (lines 518 to 538).

#6. In addition, there are concepts that are difficult to reconcile with the biology of plants and microbes in soil: (L. 353-55) how can obligate biotrophs be predicted by edaphic factors only?

Thank you for your comment. In line with the previous comment, we added that more information on plant communities would also permit to better investigate the distribution patterns of microbial biotrophs (Discussion section of the manuscript - line 463). Previous results (still valid in the present manuscript), show how biotrophs were influenced by all three types of environmental drivers. Indeed, our variation partitioning analysis (former Supplementary figure S4 and Figure 4) showed that soil properties, climate and vegetation cover explained spatial patterns of biotrophs, and not only edaphic factors.

The purpose of the concept mentioned in the previous manuscript version (lines 353-355) was to highlight that some properties impacting microbial functional groups are still poorly investigated (e.g., bulk density, carbonate content) but, based on our findings, they play a role. That reflection aimed at encouraging the inclusion of such properties in future assessments and, thus, improving our knowledge on them. Nevertheless, they are not considered as being the only influencing factors (see variation partitioning results) on the examined functional groups. We think that the current version better present our thought (lines 401 to 404).

#7. Likewise, some of the proposed actions appear impractical: (L.411-12) varying soil pH will likely alter mineral availability for plant uptake, hence the other soil variables may not be kept "fixed".

Thank you very much for your comment, we recognize the impracticability of the proposal and removed it from the Discussion (lines 421 to 450). However, we added some visualizations to better illustrate the importance of considering interactions among drivers as an informative tool (see Results lines 268 to 279 and Supplementary Fig. 16). The purpose of this additional information is to highlight that investigating combinatorial patches of climatic and soil conditions with vegetation cover may reveal different patterns between soil microbiome and environmental factors, that would be overlooked when considering factors acting in parallel. The creation of cluster areas based on such interactions may facilitate the development of customized policy-actions.

#8. The lack of access to raw sequences, sampling information as well as scripts used to produce figures and statistical analyses prevents these critical data to be properly evaluated. I suggest to release this information promptly and, whenever possible, by using open-access established repositories (e.g., ENA-EBI, Zenodo, GitHub...).

We have now provided all information requested in terms of raw data and scripts on the *Nature Communications'* Google Drive. The data used for this analysis will be made available in the European Soil Data Centre (ESDAC - <https://esdac.jrc.ec.europa.eu/>) as soon as the article will be published. Please note that, following European Commission's open access policy, ESDAC already hosts all LUCAS-related datasets: <https://esdac.jrc.ec.europa.eu/resource-type/datasets>.

Reviewers' Comments:

Reviewer #1:

Remarks to the Author:

The authors reviewed all of my feedback, which overlapped with the other reviewer's feedback (particularly the issues with putative functional assignment), but also appear to have addressed the other reviewers' unique feedback as well.

Reviewer #2:

Remarks to the Author:

The authors addressed my comments adequately and I recommend the manuscript for publication.

Reviewer #3:

Remarks to the Author:

In this revised version of the manuscript, Labouyrie and co workers addressed the majority of my criticisms in a satisfactory way. For what concerns my evaluation, I particularly appreciated the revised paragraph 'Study limitations and perspectives', providing the readers with a balanced assessment of manuscript's findings. I also appreciated the effort of making a clear reference to predicted/potential/putative functions, which accurately reflects the content of the manuscript. Likewise, the methodological part now contains the details I missed in the first place and I am glad that some of my suggestions helped in strengthening scripts/findings. As indicated in the first round of revision, I am confident this manuscript has the potential to become a very useful reference for the immediate field of research and beyond (e.g., large scale cross-microbiome surveys)-the revision has certainly strengthened this potential. What I am left with is a couple of minor suggestions

1) In light of what discuss about the nature of the findings, I'd propose to use the term 'inferred' (instead of 'associated') in the title when relating to the functions.

2) When I referred to the 'fate of technical replicates' in the first round of revision I meant the sentence 'a random selection of 12 samples were subjected to repeated analysis to validate quality'. My question still stands: where are the results of this validation? I assume an additional supplementary figure and/or table will clarify this point.

3) Data Availability. I welcome the proposition of the authors, however the only sequencing resource I could find on the ESDAC website is a reference to a published manuscript:

<https://esdac.jrc.ec.europa.eu/node/66326>

At first glance the website has no structure to host either raw sequencing data or scripts.

My former recommendation still remains:

-raw sequences: ENA

-scripts: github

-supplementary datasets (which cannot be integrated as supplementary information): zenodo

All of these external repositories can be cross-referenced to the ESDAC website. This strategy will ensure a broader access to your data.

4) A manuscript very relevant to this submission has recently been published (i.e., PMID: 36443458). I think the readers will benefit from a (brief) comparison of the works, highlighting in particular analogies/dissimilarities.

A final consideration, for which the authors are not required to take any action. Sequencing

databases (at least the ones that are regularly curated) are no static entities, e.g.
<https://www.arb-silva.de/news/view/2022/03/07/update-on-silva-taxonomy/> => that's 4 years
after your cross citation on databases' discrepancies.

#END

REVIEWER COMMENTS

Reviewer #1 (Remarks to the Author):

The authors reviewed all of my feedback, which overlapped with the other reviewer's feedback (particularly the issues with putative functional assignment), but also appear to have addressed the other reviewers' unique feedback as well.

Thank you very much for this positive feedback!

--

Reviewer #2 (Remarks to the Author):

The authors addressed my comments adequately and I recommend the manuscript for publication.

Thank you very much for this positive feedback!

--

Reviewer #3 (Remarks to the Author):

In this revised version of the manuscript, Labouyrie and co workers addressed the majority of my criticisms in a satisfactory way. For what concerns my evaluation, I particularly appreciated the revised paragraph 'Study limitations and perspectives', providing the readers with a balanced assessment of manuscript's findings. I also appreciated the effort of making a clear reference to predicted/potential/putative functions, which accurately reflects the content of the manuscript. Likewise, the methodological part now contains the details I missed in the first place and I am glad that some of my suggestions helped in strengthening scripts/findings. As indicated in the first round of revision, I am confident this manuscript has the potential to become a very useful reference for the immediate field of research and beyond (e.g., large scale cross-microbiome surveys)-the revision has certainly strengthened this potential.

Thank very much for your positive feedback and further suggestions that we integrated now to the revised manuscript and supplements.

What I am left with is a couple of minor suggestions

1) In light of what discuss about the nature of the findings, I'd propose to use the term 'inferred' (instead of 'associated') in the title when relating to the functions.

The title has been adapted accordingly, thank you.

2) When I referred to the 'fate of technical replicates' in the first round of revision I meant the sentence 'a random selection of 12 samples were subjected to repeated analysis to validate quality'. My question still stands: where are the results of this validation? I assume an additional supplementary figure and/or table will clarify this point.

Thank you for the clarification. In light of your previous comment on library-to-library variability (first revision round), we added a supplementary figure 1 to the Supplementary methods 1 (see below) and replaced the old sentence with:

"For both bacterial and fungal DNA datasets, 12 randomly selected samples were replicated during library preparation to verify that library-to-library variability was low. Data were processed as described for the full data set until normalization of the (z)OTU table (threshold 16S: 12'277 counts and ITS: 458 counts). For fungi,

two resequenced samples were excluded from the analysis, as one of the replicates in each pair had a much smaller number of reads than the threshold. In unconstrained NMDS ordination based on Bray-Curtis dissimilarity, the pairs of replicates grouped by sample ID, confirming the reproducibility of sequencing results (Fig. 1).

Fig. 1: Ordination plots for pairs of resequenced samples (replicates) for (a) bacteria and (b) fungi."

Here, rather than relying on correlations within- and between- samples, we opted for a distance-based ordination method using Bray-Curtis dissimilarity index, that better accounts for double zeros among pairs of samples in sparse data.

3) Data Availability. I welcome the proposition of the authors, however the only sequencing resource I could find on the ESDAC website is a reference to a published manuscript:

<https://esdac.jrc.ec.europa.eu/node/66326>

At first glance the website has no structure to host either raw sequencing data or scripts.

My former recommendation still remains:

- raw sequences: ENA
- scripts: github
- supplementary datasets (which cannot be integrated as supplementary information): zenodo

All of these external repositories can be cross-referenced to the ESDAC website. This strategy will ensure a broader access to your data.

Thank you for your recommendations; the raw sequences, R scripts and supplementary datasets (i.e., metadata) are now all present in a dedicated page on the European Soil Data Centre (ESDAC) and will be made publicly available as soon as the article is published (same day of publication). The page to be published will have the address: <https://esdac.jrc.ec.europa.eu/content/soil-biodiversity-dna-bacteria-and-fungi>. In particular, the bacterial and fungal raw sequences data will be available as two downloadable compressed .zip files containing .fastq files for the 885 LUCAS sites sampled for the biodiversity module. Metadata (e.g., soil properties, vegetation information, climatic variables) will be made available as well (as large Excel sheets), next to all R scripts (as .txt) used in the workflow for data analyses with a README document explaining their content. The supplementary Excel files that could not fit in the supplementary information will also be available for download.

This follows JRC's policy for open access applied to all generated datasets upon publication in peer-review journals. ESDAC already hosts datasets that are the main outputs of papers in Nature Communications. See, for instance, the data download pages for Borrelli et al. "An assessment of the global impact of 21st century land use change on soil erosion", 2017, Nature Communications: <https://esdac.jrc.ec.europa.eu/content/global-soil-erosion> and for Alewell et al. "Global phosphorus shortage will be aggravated by soil erosion", 2020, Nature Communications: <https://esdac.jrc.ec.europa.eu/content/global-phosphorus-losses-due-soil-erosion>.

As the single reference point for EU wide soil data and knowledge, ESDAC hosts all relevant soil data and information at European level. It has known a rise in number of licensed datasets hosted and an increasing interest in terms of visits and downloads over the past years (see figure below from Panagos et al. 2022 <https://bsssjournals.onlinelibrary.wiley.com/doi/full/10.1111/ejss.13315>).

There are several advantages in hosting the data in ESDAC, as it:

- allows users to download data from a single open-data source,
- permits to monitor download rates and data usage,
- ensures visibility. For instance, ESDAC communicates the publications of new datasets through a newsletter having 12,500 subscribers,
- hosts a full set of metadata and datasets (i.e., LUCAS Soil surveys) that may be associated to soil DNA information,
- provides a helpdesk that promptly responds to any questions relevant to data, scripts and published material.

We thus think that, whenever feasible, all materials related to LUCAS Soil and its biodiversity module should be hosted on ESDAC. We have now made the needed structural changes so that ESDAC dedicated page can host them.

Numbers of datasets downloads from European Soil Data Centre (ESDAC).

4) A manuscript very relevant to this submission has recently been published (i.e., PMID: 36443458). I think the readers will benefit from a (brief) comparison of the works, highlighting in particular analogies/dissimilarities.

Thank you for mentioning this reference, we have integrated how metabolomic information as investigated in Shaffer et al. could complement further outputs on metagenomics, metatranscriptomics and metaproteomics in the part of the discussion dealing with the limitations of our study (see lines 487 to 492 of the latest revised manuscript).

A final consideration, for which the authors are not required to take any action. Sequencing databases (at least the ones that are regularly curated) are no static entities, e.g. <https://www.arb-silva.de/news/view/2022/03/07/update-on-silva-taxonomy/> => that's 4 years after your cross citation on databases' discrepancies.

Thank you for this last consideration that we will keep in mind for further works.

#END